# Mechanism of SK2 channel gating and its modulation by the bee toxin apamin and small molecules

Samantha J Cassell[1†], Weiyan Li[2†], Simon Krautwald[3], Maryam Khoshouei[4], Yan Tony Lee[2], Joyce Hou[2], Wendy Guan[2], Stefan Peukert[3], Wilhelm Weihofen[1], Jonathan R Whicher[1]*

[1]Discovery Sciences, Novartis Biomedical Research, Cambridge, United States; [2]Cardiovascular and Metabolism, Novartis Biomedical Research, Basel, Switzerland; [3]Global Discovery Chemistry, Novartis Biomedical Research, Cambridge, United States; [4]Discovery Sciences, Novartis Biomedical Research, Basel, Switzerland

*For correspondence:
jonathan.whicher@novartis.com

†These authors contributed equally to this work

## eLife Assessment

In this **important** manuscript, Cassell and colleagues set out on a mechanistic and pharmacological exploration of an engineered chimeric small conductance calcium-activated potassium channel 2 (SK2). They show **compelling** evidence that the SK2 channel possesses a unique extracellular structure that modulates the conductivity of the selectivity filter, and that this structure is the target for the SK2 inhibitor apamin. The interpretations are sound and the writing is clear, and the manuscript was strengthened during review by providing more detailed information for the electrophysiological experiments and the structural analyses attempted, in addition to relating dilation of the filter to mechanisms of inactivation in other potassium channels. This high-quality study will be of interest to membrane protein structural biologists, ion channel biophysicists, and chemical biologists, and will help to inform future drug development targeting SK channels.

**Abstract** Small-conductance calcium-activated potassium channel 2 (SK2) serves a variety of biological functions by coupling intracellular calcium dynamics with membrane potential. SK2 modulators are in development for the treatment of neurological and cardiovascular diseases, though the mechanisms of pharmacological modulation remain incompletely understood. We determined structures of an SK2–4 chimeric channel in $Ca^{2+}$-bound and $Ca^{2+}$-free conformations and in complex with the bee toxin apamin, a small molecule inhibitor, and a small molecule activator. The structures revealed that the S3–S4 linker forms a hydrophobic constriction at the extracellular opening of the pore. Apamin binds to this extracellular constriction and blocks the exit of potassium ions. Furthermore, we identified a structurally related SK2 inhibitor and activator that bind to the transmembrane domains. The compounds exert opposing effects on gating by differentially modulating the conformation of the S6 helices. These results provide important mechanistic insights to facilitate the development of targeted SK2 channel therapeutics.

## Introduction

Fundamental to a variety of biological processes, ion channels are an important class of therapeutical targets. Small-conductance calcium-activated potassium (KCa2.x, or SK1, 2, and 3) channels are activated by increased intracellular $Ca^{2+}$ to induce potassium efflux and regulate membrane potential (*Köhler et al., 1996*; *Bond et al., 2004*; *Blatz and Magleby, 1986*). In this role, SK1, 2, and 3 mediate

cellular excitability and have different but overlapping functions in many cell types including neurons, endothelial cells, and cardiomyocytes (*Adelman et al., 2012*). In particular, the SK2 channel regulates synaptic transmission and plasticity, learning and memory, and cardiac action potentials and thus has attracted attention as a potential target for the treatment of neurological and cardiovascular diseases (*Bond et al., 2004*; *Hammond et al., 2006*; *Zhang et al., 2008*). SK2 activators reduce cellular excitability and are potential therapeutics for alcohol dependence (*Hopf et al., 2011*), ataxia (*Alviña and Khodakhah, 2010*), epilepsy (*Anderson et al., 2006*), and stroke (*Allen et al., 2011*). Conversely, SK2 inhibitors increase cellular excitability and have been proposed for the treatment of Alzheimer's disease (*Proulx et al., 2015*) and atrial fibrillation (*Diness et al., 2011*).

The cryo-EM structure of the related SK4 (KCa3.1, IK) channel provided the first insights into the architecture and mechanism of $Ca^{2+}$-dependent gating for the SK channel family (*Lee and MacKinnon, 2018*). SK4 channels form non-domain swapped tetramers, with each subunit containing six transmembrane helices S1 to S6 (*Köhler et al., 1996*; *Lee and MacKinnon, 2018*). S1–S4 form a voltage-sensor like domain where the S4 helices lack the positively charged residues necessary for voltage sensitivity. The S5 and S6 helices form the potassium pore. Within the potassium pore lies the selectivity filter, a structure unique to potassium channels that is required for rapid and selective conductance of $K^+$ ions (*Doyle et al., 1998*). Following the S6 helices, there are two intracellular helices (HA and HB) that form the binding site for the $Ca^{2+}$-binding protein calmodulin (CaM), which acts as the $Ca^{2+}$ sensor to gate SK channels (*Xia et al., 1998*). The CaM C-lobe is constitutively bound to the HA and HB helices, and upon an increase in intracellular $Ca^{2+}$, the CaM N-lobe binds to a unique S4–S5 linker, inducing a conformational change in the S6 helices to open the potassium pore and activate the channel (*Lee and MacKinnon, 2018*).

SK2 activators described to date bind at the interface of the CaM N-lobe and S4–S5 linker and function by stabilizing this interaction (*Lee and MacKinnon, 2018*; *Brown et al., 2020*; *Shim et al., 2019*). On the other hand, known SK2 inhibitors target the extracellular and/or transmembrane regions and were proposed to function by either direct pore block or negative gating modulation (*Brown et al., 2020*). The binding site for the bee venom toxin apamin, a cyclic 18-residue peptide inhibitor, has been mapped to the extracellular loop regions of SK2 (*Nolting et al., 2007*; *Weatherall et al., 2011*; *Lamy et al., 2010*). Apamin is of historical importance as it was used to elucidate the physiological role of SK2 and apamin inhibition of SK2 increases neuronal excitability and improves learning and memory (*Blatz and Magleby, 1986*; *Messier et al., 1991*). Apamin inhibits SK1, 2, and 3 but not 4 and is most potent against SK2 with an $IC_{50}$ of ~70 pM (*Köhler et al., 1996*; *Kuzmenkov et al., 2022*). Functional mutagenesis of apamin identified two arginine residues that are essential for inhibition (*Vincent et al., 1975*). Mutagenesis experiments on the SK2 channel indicated that residues in the extracellular loops between S3 and S4 (S3–S4 linker) and between S5 and S6 are important for apamin binding and inhibition (*Nolting et al., 2007*; *Weatherall et al., 2011*; *Lamy et al., 2010*). Since the S3–S4 linker is predicted to be distant to the pore, an allosteric mechanism of apamin inhibition rather than a direct pore block has been suggested (*Lamy et al., 2010*). Attempts to recapitulate the potency and selectivity of apamin with small molecules resulted in the development of a class of inhibitors, such as UCL1684, that are predicted to have an overlapping binding site with apamin (*Ishii et al., 1997*; *Castle et al., 1993*; *Chen et al., 2000*). These small molecules generally carry two positive charges, which may mimic the two arginine residues in apamin that are essential for inhibition (*Vincent et al., 1975*). Further characterization of the molecular mechanisms of inhibition by apamin and the small molecule pore blockers is required to understand how the interaction between the S3–S4 linker and the essential arginine residues/positive charges block ion conduction in SK2.

Another class of small molecule SK2 inhibitors acts as negative gating modulators by shifting the $Ca^{2+}$ dependence of activation to higher $Ca^{2+}$ concentrations (*Jenkins et al., 2011*; *Simó-Vicens et al., 2017*). One such inhibitor, AP31969, is currently in clinical trials for the treatment of arrhythmia (*Saljic et al., 2024*). Mutagenesis experiments with the structurally related inhibitor AP14145 suggest that these compounds bind within the pore directly below the selectivity filter (*Simó-Vicens et al., 2017*). Similar potencies on SK1, 2, and 3 channels have been reported most likely due to the homology of pore lining residues in the SK family. However, selective SK1 inhibitors were developed that take advantage of a unique residue, Ser293, on S5 (*Hougaard et al., 2012*). Interestingly, only small modifications to this family of inhibitors are sufficient to switch the activity profile from inhibition to activation of SK1. However, it remains unclear how compounds that bind the transmembrane regions of

SK channels affect $Ca^{2+}$-dependent gating, which is driven by the interaction between CaM and the intracellular domains.

Despite SK2 being a prominent therapeutic target for both neurological and cardiovascular diseases, no structure of human SK2 has been reported to date. Although the structure of rat SK2 was reported while this manuscript was in preparation (*Nam et al., 2025*). To enable high-resolution cryo-EM studies of human SK2, we designed a chimera (SK2–4) that contains the transmembrane and extracellular domains of human SK2 and intracellular domains of human SK4. The structures of the SK2–4/CaM complexes in the $Ca^{2+}$-bound and $Ca^{2+}$-free conformations demonstrate that SK2 and SK4 adopt similar overall architectures and share a similar mechanism for $Ca^{2+}$-dependent gating. However, unlike SK4, we observed a structured S3–S4 linker that induces a conformational change in the selectivity filter and forms a hydrophobic constriction at the extracellular opening of the SK2 pore. Apamin binds to the extracellular constriction formed by the S3–S4 linker to block potassium efflux. In addition, high-throughput screening and medicinal chemistry optimization efforts yielded a new class of potent SK2 inhibitors that bind to a novel pocket formed by the S5, S6, and pore helices and induce closure of the S6 helices. Structure-guided design efforts enabled switching the activity profile toward activation while retaining the same binding mode. The detailed understanding of two distinct mechanisms of SK2 channel inhibition, extracellular pore block and negative gating modulation, and a new mechanism for channel activation presented here should facilitate the rational design of potent and selective SK2 modulators.

## Results

### Characterization of the SK2–4 chimeric channel

We attempted to solve the structure of wild-type (WT) human SK2 (*Figure 1—figure supplement 1*) expressed and purified from mammalian cells in the presence of CaM and $Ca^{2+}$. However, 2D class averages showed disorder in the intracellular region (*Figure 2—figure supplement 1A*). Subsequent 3D reconstruction generated a low-resolution, anisotropic map for the transmembrane helices and no interpretable density for the intracellular domains.

The SK4 channel cryo-EM structure suggested that the intracellular regions of SK4 may be more rigid than those of SK2 (*Lee and MacKinnon, 2018*). We hypothesized that the SK4 intracellular domains would remain ordered when fused to the SK2 TM domains. A chimeric construct was created with the goal of preserving all transmembrane and extracellular regions of human SK2 while replacing the N- and C-terminal intracellular regions with those of human SK4 (*Figure 1A*, *Figure 1—figure supplement 1*, and methods for details). Within the intracellular regions, only the N-terminal portion of the HA helix (residues 401–412 in SK2) that was predicted to interact with the S4–S5 linker retained the SK2 sequence (*Lee and MacKinnon, 2018*).

To confirm that the SK2–SK4 chimeric channel (SK2–4) behaves similarly to the WT SK2 channel, we compared the function and pharmacology of SK2–4 overexpressed in CHO cells to WT human SK2 and SK4 using an automated electrophysiology instrument, the Qube (Sophion). SK currents were recorded in the 'whole-cell' configuration under voltage clamp where a saturating concentration of a specific SK2 inhibitor was applied at the end of experiment to isolate SK current. Electrophysiological measurements demonstrated that SK2–4 current has a similar reversal potential as WT SK2 and SK4 currents (*Figure 1B*), which is near the expected value for a $K^+$-selective channel based on the $K^+$ concentration gradient. To evaluate $Ca^{2+}$-dependent activation, we measured average SK current amplitude from different cell populations each exposed to a different intracellular free $Ca^{2+}$ concentration ranging from 0.1 to 20 µM (i.e., each cell population was only tested with an individual free intracellular $Ca^{2+}$ concentration). Like WT SK2 and SK4, the activation of SK2–4 is dependent on intracellular $Ca^{2+}$ concentration (*Figure 1C*). Despite the potential limit in diffusion with the 'whole-cell' configuration which precludes the ability to fully control the intracellular free $Ca^{2+}$ concentration, our results clearly showed that the $Ca^{2+}$ sensitivity of SK2–4 is higher than that of SK2 but similar to SK4, consistent with the CaM-binding domain of the chimera stemming from SK4. In addition, SK2–4 activity was inhibited by known SK2 inhibitors apamin and AP14145 with an $IC_{50}$ of 0.7 nM and 2.4 µM, respectively (*Figure 1D*). Of note, neither of these inhibitors is active against the WT SK4 channel, indicating that key regions of the SK2 native structure are preserved in the chimera (*Kuzmenkov et al., 2022*; *Simó-Vicens et al., 2017*). We also tested dozens of other available SK2 inhibitors

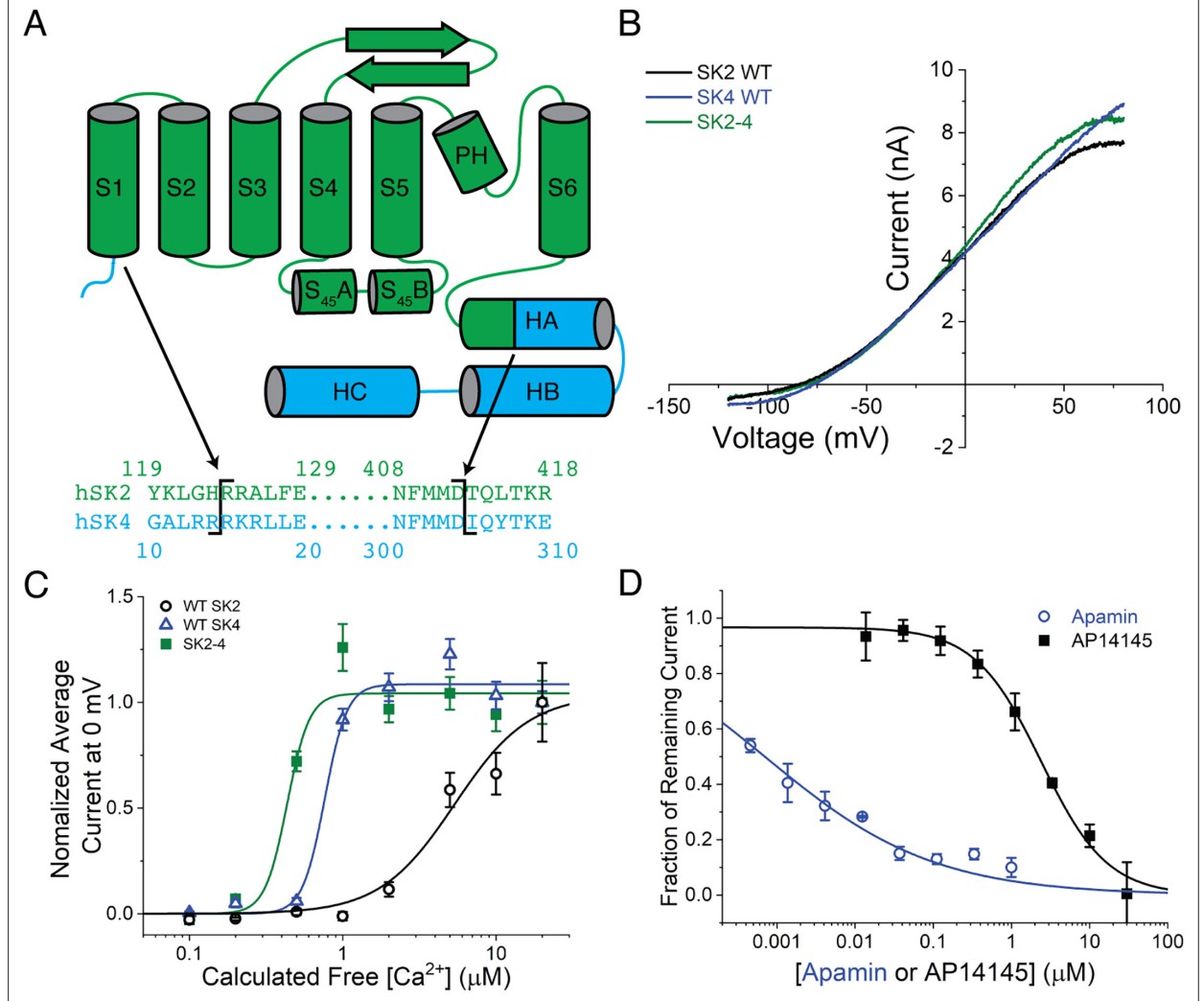

**Figure 1.** Design and characterization of the SK2–4 chimera. (**A**) Cartoon representation of the SK2–4 chimera. Residue numbers in the SK2 regions of the structure correspond to the human SK2 sequence described by **Desai et al., 2000** and the SK4 regions of the structure are numbered sequentially from the SK2 regions. The sequence alignment of human SK2 (green) and human SK4 (cyan) indicates the chimera boundaries (black lines). (**B**) Representative current traces recorded with the Qube instrument in response to a ramp voltage protocol from –120 to 80 mV show that reversal potentials of WT SK2 (black), SK4 (blue), and SK2–4 (green) currents are all around –85 mV demonstrating similar potassium selectivity. The shown SK current traces are isolated by subtracting leak current determined under saturating concentration of inhibitor (UCL1684 or TRAM-34). Intracellular solution contains 2 μM $Ca^{2+}$ based on calculation using Maxchelator (see methods). (**C**) SK2–4 (green squares) is activated by increasing intracellular $Ca^{2+}$ like WT SK2 (black circles) and SK4 channels (blue triangles). For each channel type, average SK current amplitudes from different cell populations each exposed to a different intracellular free $Ca^{2+}$ concentration (calculated) ranging from 0.1 to 20 μM were normalized to average value at 20 μM $Ca^{2+}$. Apparent $EC_{50}$s for SK2, SK4, and SK2–4 are 5.4, 0.8 and 0.4 μM, respectively. Note that actual intracellular free $[Ca^{2+}]$ might differ from intracellular buffer due to the limit of diffusion in whole-cell configuration. Data points reflect mean ± SEM (n = 24). (**D**) SK2–4 is inhibited by apamin (blue circles, $IC_{50}$ = 0.7 nM), which targets the extracellular domains of SK2, and AP14145 (black squares, $IC_{50}$ = 2.4 μM), which targets the transmembrane domains of SK2. Data points reflect mean ± SD (n = 4). Change in SK2–4 current amplitude for different cell populations each exposed to a different extracellular compound concentration was normalized to DMSO control to construct dose responses.

The online version of this article includes the following figure supplement(s) for figure 1:

**Figure supplement 1.** Sequence alignment of human SK1, SK2, SK3, and SK4.

of diverse scaffolds and observed a close correlation between potency on WT SK2 and the SK2–4 chimera (data not shown). Given the high degree of agreement between our characterization and previously reported data, we are confident that SK2–4 recapitulates the native activity and transmembrane architecture of WT SK2.

## Structure of SK2–4/CaM in Ca²⁺-bound and Ca²⁺-free states

SK2–4 was co-expressed with CaM and purified from mammalian cells in the presence of Ca²⁺. The cryo-EM structure was determined to a final resolution of 3.1 Å with most regions of the structure well-defined by density apart from the C-terminal HC helical bundle (*Figure 2A*, *Figure 2—figure supplements 1 and 2A*, *Supplementary file 1*). Residue numbers in the SK2 regions of the structure correspond to the human SK2 sequence described by *Desai et al., 2000* and the SK4 regions of the structure are numbered sequentially from the SK2 regions (*Figure 1A*).

SK2–4/CaM forms a non-domain swapped tetramer with four CaM molecules bound, similar to the previously reported SK4/CaM structure (*Lee and MacKinnon, 2018*; *Figure 2A*). In the presence of Ca²⁺, the C-lobe of each CaM is bound to the HA and HB helices and the N-lobe is bound to a short helix in the S4–S5 linker, $S_{45}A$, on the neighboring subunit (*Figure 2—figure supplement 1G, H*). The measured SK2–4 pore radius (3.8 Å) indicates that the S6 helices are in an open conformation in the presence of Ca²⁺ (*Figure 2C, D*).

We also purified SK2–4/CaM in the absence of Ca²⁺ and determined the cryo-EM structure at 3.4 Å (*Figure 2B*, *Figure 2—figure supplements 1 and 2B*, *Supplementary file 1*). In the structure, there is clear density for the S1–S6 regions, but as observed in the closed conformation of SK4, the CaM density is weak and the N-lobes of CaM are dissociated from the $S_{45}A$ helices (*Lee and MacKinnon, 2018*). Furthermore, the $S_{45}B$ and S6 helices collapsed around the pore axis, and the pore is closed with a radius of 1 Å at the intracellular gate at Val390 (*Figure 2C, D*). The structures of SK2–4/CaM in the Ca²⁺-bound and Ca²⁺-free conformations confirm that the chimeric channel is gated by Ca²⁺ and that the mechanism of channel gating is similar to that of SK4.

## S3–S4 linker structure

Although SK2–4 and SK4 share similar overall architecture, there are notable differences in the SK2 portion spanning the S1–S6 transmembrane region. The extracellular S3–S4 linker is not visible in the structure of SK4, indicating flexibility and lack of defined structure (*Lee and MacKinnon, 2018*). In contrast, the S3–S4 linker of SK2–4 is well-defined by density in both the Ca²⁺-bound and Ca²⁺-free structures and forms a two-stranded anti-parallel β-turn (residues Gly231–Asp253) that extends over the S5 and S6 helices (*Figure 2—figure supplement 2A, B*, *Figure 3*). The β-turn interacts with residues Tyr335 and His336 at the C-terminus of the S5 helix (*Figure 3B*). His336 forms an edge-to-face interaction with Trp237 and a hydrogen bond with Ser248, whereas Tyr335 forms a hydrogen bond with Asp253. The β-turn features an eight amino acid loop (residues 240–247) that extends into the potassium pore (*Figures 3C and 4A*, *Figure 4—figure supplement 1A, B*). Arg240, Phe243, and Tyr245 within this loop form direct interactions with residues at the C-terminus of the selectivity filter. Arg240 and Tyr245 form a salt bridge and hydrogen bond with the side chain and backbone of Asp363 from a neighboring subunit, respectively. Phe243 extends into the ion conduction path and is within C–H/O bonding distance to the backbone carbonyl of Gly362 (*Figure 3C*). In this position, the four Phe243 residues form edge-to-face interactions that create a hydrophobic constriction at the extracellular opening of the pore with a radius of 1.8 Å, which is expected to prohibit efflux of hydrated K⁺ ions measuring 6 Å in diameter (*Figures 2D, 3A, and 4A*). Therefore, even though the intracellular gate at Val390 is open in the SK2–4/CaM Ca²⁺-bound structure, the observed conformation is expected to be non-conductive. For conduction to occur, a structural rearrangement of the S3–S4 linker that increases the diameter of the extracellular constriction would be required.

Focused 3D classification of the S3–S4 linker was unsuccessful in identifying particle subsets with a dilated extracellular constriction, suggesting that either none or a very small percentage of Ca²⁺-bound SK2–4 is in a conductive state. This, along with the similar conformation of the S3–S4 linker in the Ca²⁺-bound and Ca²⁺-free states of SK2–4, suggests that Ca²⁺-dependent intracellular gate dynamics are not coupled to the conformation of the S3–S4 linker. Other yet-to-be-identified physiological factors are likely required to dilate the extracellular constriction.

All residues participating in interactions between the S3–S4 linker and the pore resides are fully conserved in SK1, 2, and 3 (*Figure 1—figure supplement 1*), suggesting that S3–S4 linker conformation is an important structural feature of the SK channels. In addition, the S3–S4 linker conformation is important for apamin inhibition as mutation of Tyr245 and His336, which are involved in hydrogen-bonding interactions between the S3–S4 linker and pore residues, reduce apamin potency (*Weatherall et al., 2011*; *Lamy et al., 2010*).

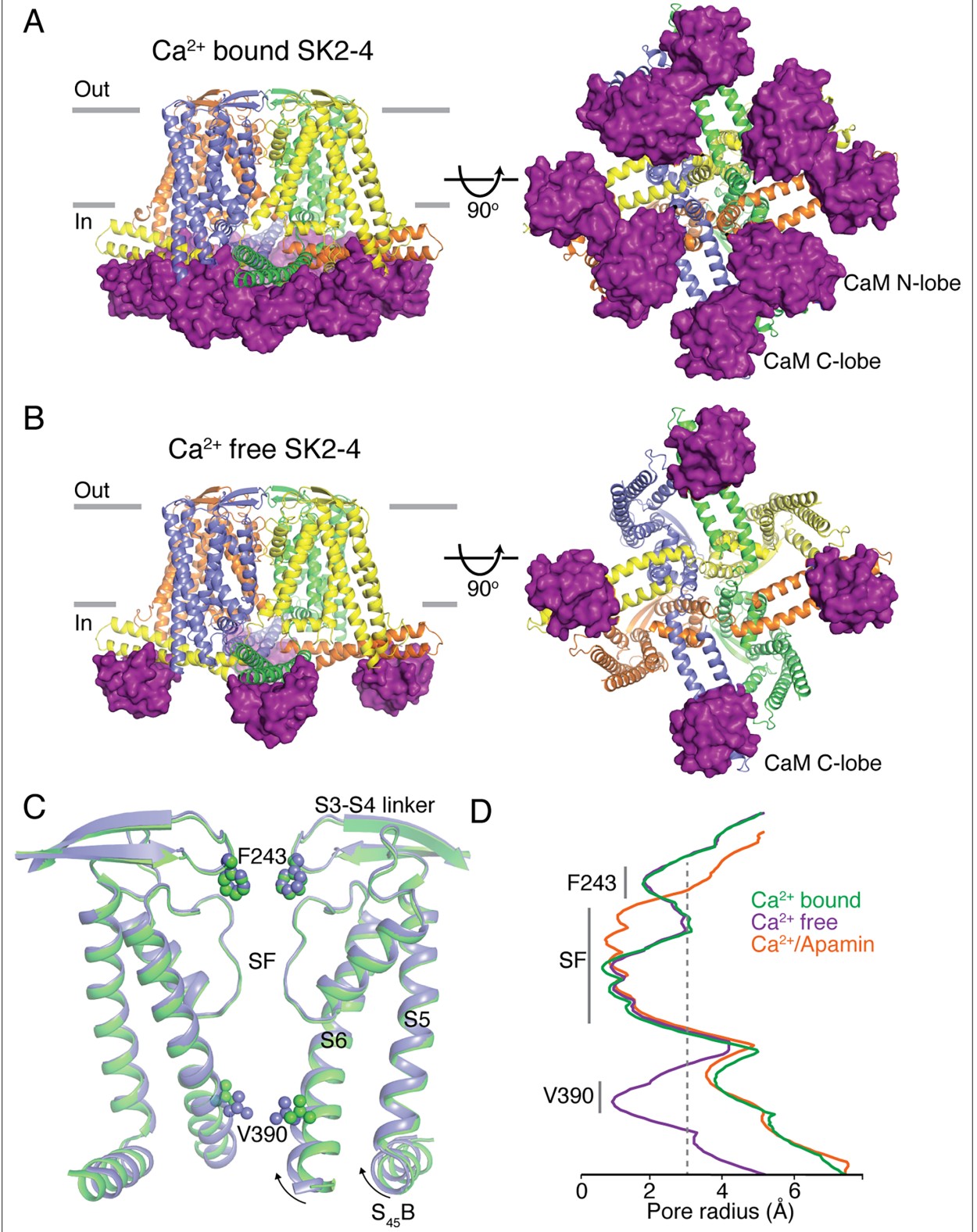

**Figure 2.** SK2–4 chimera architecture. Structures of Ca²⁺-bound (**A**) and Ca²⁺-free (**B**) SK2–4. SK2–4 is shown as a cartoon with each subunit of the tetramer in a different color. The gray lines (left) indicate membrane boundaries. In Ca²⁺-bound SK2–4 (**A**) both the N- and C-lobes of CaM (purple surface) are associated with the intracellular domains. In Ca²⁺-free SK2–4 (**B**) only the CaM C-lobe is bound (purple surface) and the N-lobe is dissociated (not shown). (**C**) Overlay of the K⁺ pore from Ca²⁺-bound (green) and Ca²⁺-free (lavender) SK2–4 structures. Same view as left panels of (**A**) and (**B**) but

*Figure 2 continued on next page*

*Figure 2 continued*

only 2 subunits are shown. (**D**) Pore radii of Ca²⁺-bound (green), Ca²⁺-free (lavender), and apamin-bound (orange) SK2–4. The location of the intracellular gate (Val390), selectivity filter (SF), and extracellular constriction (Phe243) is indicated. Gray dashed line indicates the radius of hydrated K⁺ (3 Å).

The online version of this article includes the following figure supplement(s) for figure 2:

**Figure supplement 1.** Structure determination of Ca²⁺-bound SK2–4/CaM and Ca²⁺-free SK2–4/CaM.

**Figure supplement 2.** Representative cryo-EM densities.

## Selectivity filter conformation

The selectivity filter is a crucial component of all selective K⁺ channels, including SK4. It contains the highly conserved (T/S)XG(Y/F)G motif, where the T/S hydroxyl and the backbone carbonyls of subsequent residues form four consecutive K⁺ coordination sites that perfectly mimic the hydration shell of a K⁺ ion (*Figure 4B, C*; *Doyle et al., 1998*; *Zhou et al., 2001*). Extensive structural and mechanistic studies demonstrated that this selectivity filter structure is required for the fast and selective conduction of K⁺ ions (*Morais-Cabral et al., 2001*; *Lockless, 2015*; *Shi et al., 2006*; *Derebe et al., 2011*; *Sauer et al., 2011*). Reducing the number of K⁺ coordination sites in the selectivity filter produces channels that are either not K⁺ selective, like HCN or NaK, or not conductive, like voltage-gated potassium channels (Kᵥ) that have undergone C-type inactivation (*Figure 4E–H*; *Shi et al., 2006*; *Lee and MacKinnon, 2017*; *Tyagi et al., 2022*; *Selvakumar et al., 2022*).

In the structures of SK2–4/CaM in both the Ca²⁺-bound and Ca²⁺-free conformations, Tyr361, located in the center of the selectivity filter, is rotated approximately 180° when compared with the homologous Tyr253 in SK4 (*Figure 4A, C, D*, *Figure 4—figure supplement 1A, B*; *Lee and MacKinnon, 2018*). In SK4, Tyr253 hydrogen bonds with a pore helix threonine (Thr247), and the homologous interaction is not observed in SK2–4 due to the rotation of Tyr361. In addition, the backbone carbonyls of Gly360 and Tyr361, which typically form the extracellular K⁺ coordination sites one and two, are rotated away from the center of the selectivity filter and the ion conduction path. As a result, only the two intracellular K⁺ coordination sites, three and four, are intact and occupied by K⁺. The selectivity filter of SK2–4 resembles that of HCN in both the position of Tyr361 and the number of

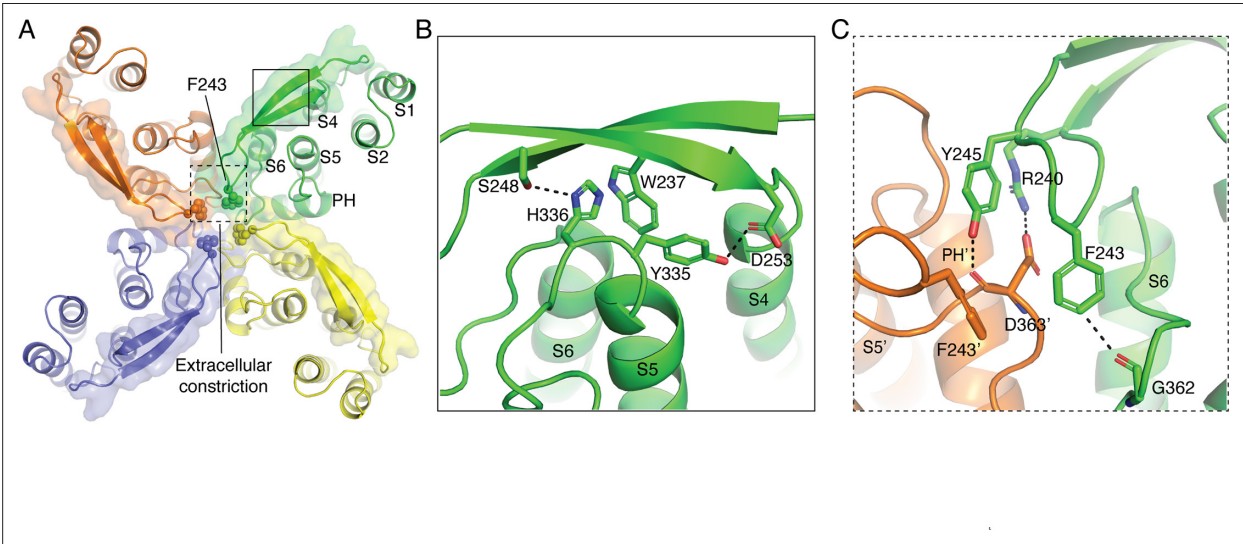

**Figure 3.** S3–S4 loop architecture. (**A**) Extracellular view of Ca²⁺-bound SK2–4 with each subunit of the tetramer in a different color. The S3–S4 linker (surface and cartoon) extends over the S5 and S6 helices. Phe243 residues (spheres) form an extracellular constriction with a radius of 1.8 Å. Boxes indicate location of the interactions shown in (**B**) and (**C**). (**B**) Interactions between the S3–S4 linker and the C-terminus of S5. His336 forms an edge-to-face interaction with Trp237 and hydrogen bond with Ser248 (dashed lines). Tyr335 forms a hydrogen bond with D253 (dashed lines). (**C**) Interactions between the S3–S4 linker and the C-terminus of the selectivity filter. Arg240 and Tyr245 (green sticks) from the S3–S4 linker form a salt bridge and hydrogen bond (dashed lines) with the side chain and backbone carbonyl of Asp363 from the neighboring subunit (orange sticks), respectively. Phe243 (green sticks) forms an edge-to-face interaction with the neighboring Phe243 (orange sticks) and is in position to form a C–H/O interaction (dashed line) with Gly262 (green sticks) from the same subunit.

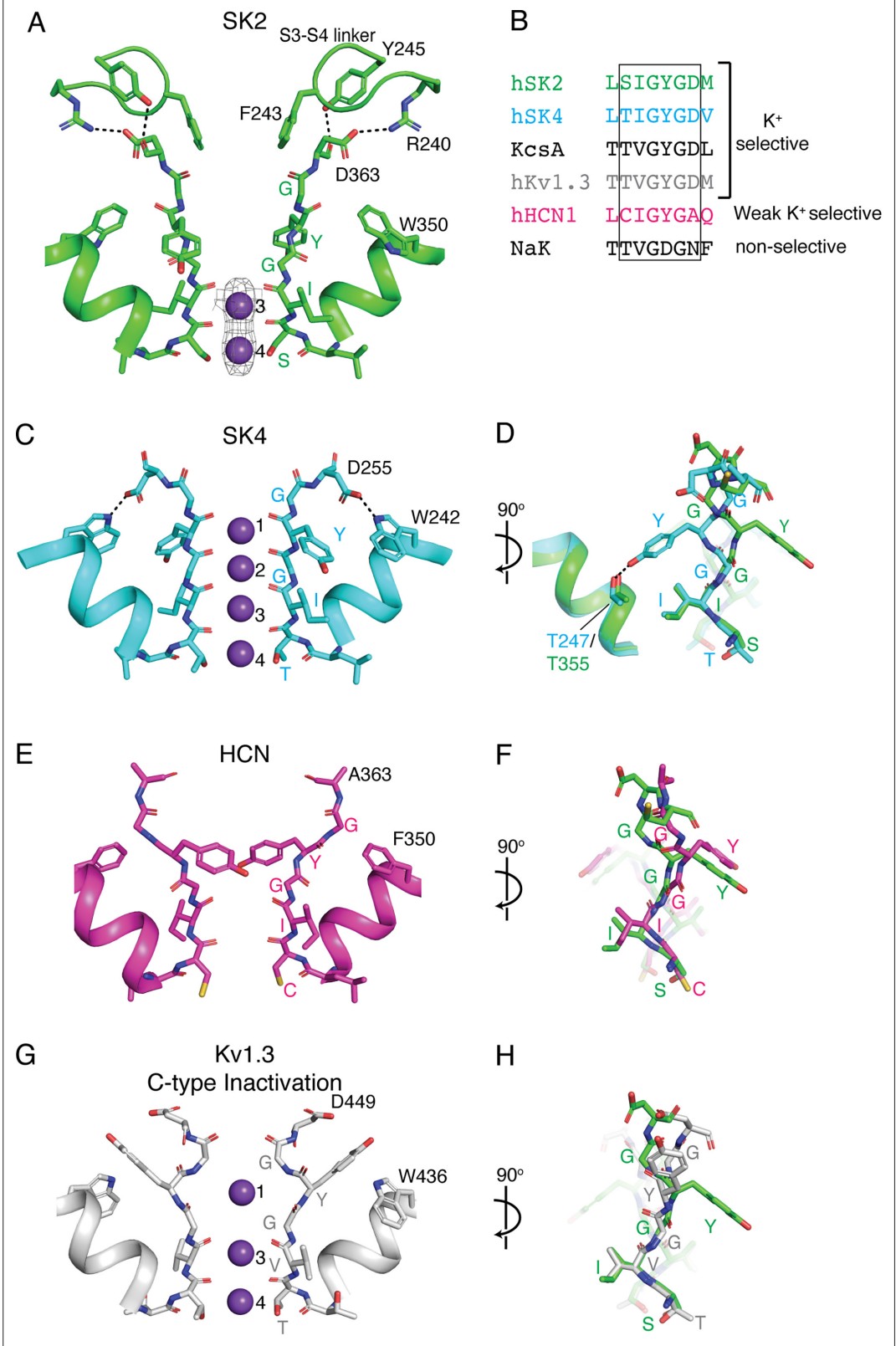

**Figure 4.** Selectivity filter conformation of SK2–4. (**A**) Ca²⁺-bound SK2–4 selectivity filter structure (green). Arg240 and Tyr245 from the S3–S4 linker form a salt bridge and hydrogen bond (dashed lines) with Asp363 and the selectivity filter adopts a conformation with two K⁺ coordination sites (purple spheres, density shown at 5 σ). (**B**) Sequence alignment of selectivity filter from K⁺ selective channels (hSK2, hSK4, KcsA, and hKv1.3) and non-

*Figure 4 continued on next page*

*Figure 4 continued*

selective cation channels (hHCN and NaK). (**C**) Structure of hSK4 selectivity filter (cyan, PDB: 6CNN). In SK4, there is a conserved hydrogen bond between the selectivity filter Asp255 and pore helix Trp242 (dashed lines), Tyr253 hydrogen bonds with Thr247, and the selectivity filter conformation creates four occupied K+ coordination sites (purple spheres). (**D**) Overlay of the SK2–4 (green) and the SK4 (cyan) selectivity filter with a 90° rotation from (**C**). (**E**) Structure of HCN selectivity filter (magenta, PDB: 5U6O). In HCN, there is no interaction between the Ala363 at the C-terminus of the selectivity filter and the pore helix Phe350. Tyr361 is in a similar conformation to Ca$^{2+}$-bound SK2–4, and the selectivity filter adopts a conformation with two ion coordination sites (the authors did not model the ions but cryo-EM density indicates two coordination sites). (**F**) Overlay of the SK2–4 (green) and the HCN (magenta) selectivity filter with a 90° rotation from (**E**). (**G**) Structure of Kv1.3 selectivity filter in a C-type inactivated conformation (magenta, PDB: 7WF3). In this conformation, there is no interaction between Asp449 at the C-terminus of the selectivity filter and the pore helix Trp436, Tyr447 is directed extracellularly, and the selectivity filter adopts a conformation with three occupied K+ coordination sites (purple spheres). (**H**) Overlay of the SK2–4 (green) and the Kv1.3 (gray) selectivity filter with a 90° rotation from (**G**).

The online version of this article includes the following figure supplement(s) for figure 4:

**Figure supplement 1.** Cryo-EM density for the SK2–4 selectivity filter.

---

ion coordination sites (*Figure 4E–H*; *Shi et al., 2006*; *Lee and MacKinnon, 2017*; *Tyagi et al., 2022*; *Selvakumar et al., 2022*).

SK2 and SK4 selectivity filters were predicted to adopt similar conformations based on sequence conservation in the selectivity filter and surrounding residues (*Figure 1—figure supplement 1*). Therefore, the conformation of the selectivity filter observed in the SK2–4 structure is likely due to the S3–S4 linker, which is disordered in SK4. As discussed above, Asp363 at the C-terminus of the selectivity filter is directed toward the extracellular S3–S4 linker and interacts with Arg240 and Tyr245 (*Figures 3C and 4A*, *Figure 4—figure supplement 1A, B*). In SK4, the homologous aspartate (Asp255) is directed toward the pore helix and forms a hydrogen bond with a conserved tryptophan (Trp242) (*Figure 4C*; *Lee and MacKinnon, 2018*). The homologous pore helix tryptophan in SK2, Trp350, adopts a different rotamer and is unable to form a hydrogen bond with Asp363 (*Figure 2—figure supplement 2A, B*, *Figure 4A*). The hydrogen-bonding interaction between a pore helix residue (usually Trp or Tyr) and the residue at the C-terminus of the selectivity filter (usually Asp or Asn) is a conserved feature of K+ selective channels (*Sauer et al., 2011*). Furthermore, inserting this interaction into non-selective channels produces K+ selective channels with four K+ coordination sites, while deletion of this interaction from K+ selective channels produces non-selective channels with two K+ coordination sites (*Derebe et al., 2011*; *Sauer et al., 2011*). Therefore, we propose that by interacting with Asp363 and thereby preventing its interaction with the pore helix Trp350, the S3–S4 linker induces the selectivity filter conformation observed in SK2–4 with two K+ coordination sites. We predict that the SK1 and 3 selectivity filters could adopt similar conformations with two K+ coordination sites because the S3–S4 linker residues that interact with Asp363 are conserved among these channels (*Figure 1—figure supplement 1*).

Except for one report indicating significant sodium permeability in rat SK2 (*Shin et al., 2005*), studies show that SK2 is K+ selective (*Köhler et al., 1996*; *Desai et al., 2000*; *Park, 1994*; *Jäger et al., 2000*). The reversal potential of SK2 and SK2–4 presented here confirms K+ selectivity (*Figure 1B*). This contrasts with the SK2–4 structures as selectivity filters with two K+ coordination sites are predicted to be non-selective (*Shi et al., 2006*; *Derebe et al., 2011*; *Sauer et al., 2011*; *Lee and MacKinnon, 2017*). However, the Ca$^{2+}$-bound SK2–4/CaM structure is in a non-conductive conformation, despite the open intracellular Val390 gate, due to Phe243 of the S3–S4 linker forming a constriction at the extracellular opening that prevents K+ efflux. Thus, the selectivity filter conformation should not affect the K+ selectivity observed in the experimental data (*Köhler et al., 1996*; *Desai et al., 2000*; *Park, 1994*; *Jäger et al., 2000*).

## Mechanism of apamin inhibition

Extensive mutational studies suggest that the SK2 S3–S4 linker is essential for apamin binding and inhibition, although the nature of their interaction remains unknown (*Nolting et al., 2007*; *Weatherall et al., 2011*). To understand how apamin interacts with the unique conformation of the S3–S4 linker and how that interaction inhibits K+ conduction, we incubated Ca$^{2+}$-bound SK2–4/CaM with excess

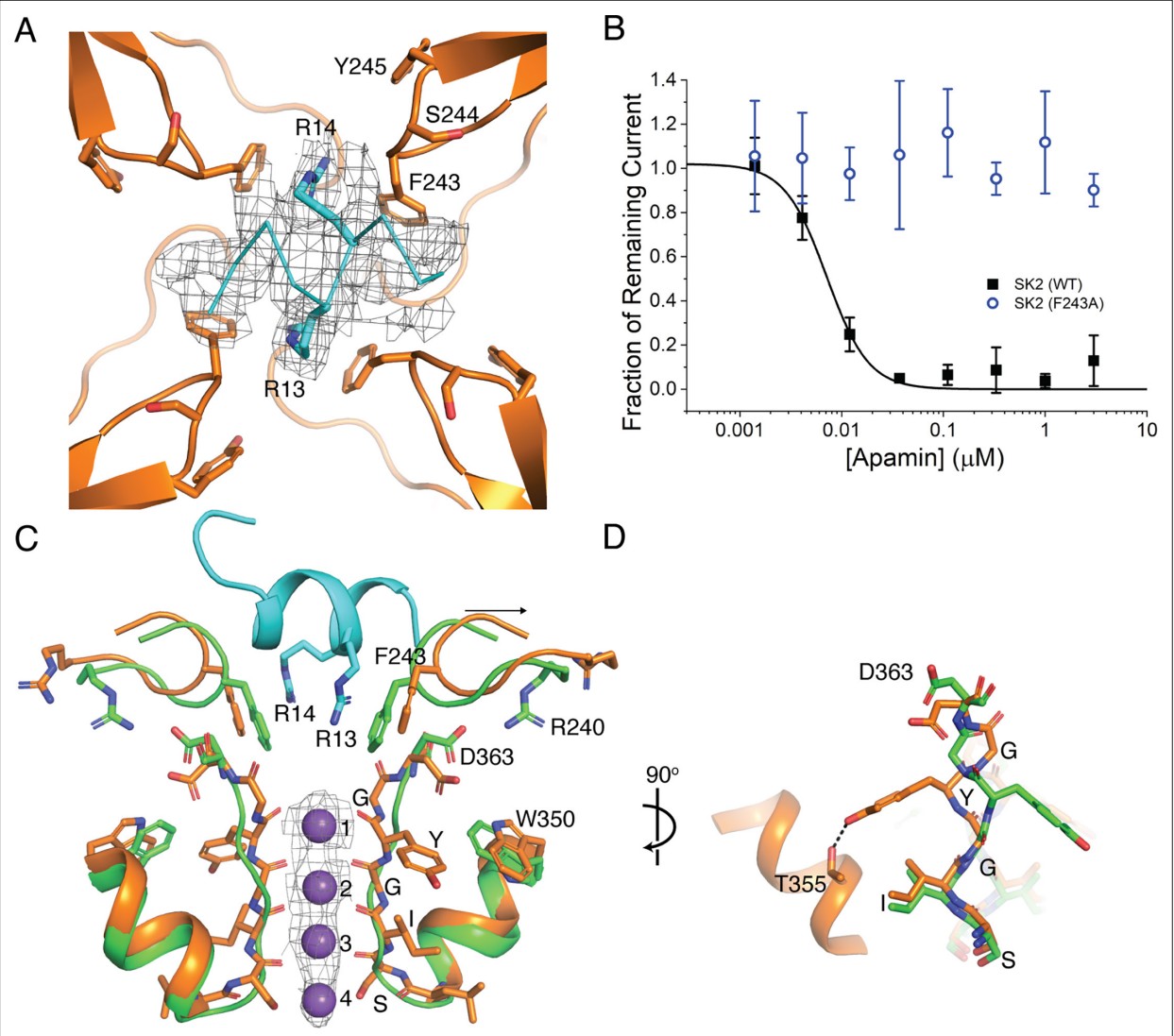

**Figure 5.** Mechanism of apamin inhibition. (**A**) Extracellular view of apamin-binding site. Apamin (cyan ribbon, density shown at 5 σ) binds to the S3–S4 extracellular gate. Apamin Arg13 and Arg14 (cyan) and the S3–S4 linker residues (orange) that surround the apamin-binding site are shown as sticks. (**B**) Apamin sensitivity of WT SK2 (black squares) and SK2 F243A (blue circles). Change in SK current amplitude for different cell populations each exposed to a different extracellular apamin concentration was normalized to DMSO control to construct dose responses. Data points represent mean ± SD (*n* = 6). The F243A mutant is insensitive to apamin up to 3 µM. (**C**) Overlay of Ca²⁺-bound SK2–4 selectivity filter (green) and apamin-bound SK2–4 selectivity filter (orange, selectivity filter shown as sticks and S3–S4 linker and pore helix shown as a cartoon). Upon apamin (cyan) binding, the S3–S4 linker retracts from the pore axis (black arrow). Asp363 reorients into hydrogen-bonding position with a rotated Trp350 side chain, Tyr361 in the selectivity filter is rotated 180° to form a hydrogen bond with Thr355 in the pore helix, and the selectivity filter adopts a conformation with four K⁺ coordination sites (purple spheres, density shown at 5 σ). (**D**) 90° degree rotation of (**C**).

The online version of this article includes the following figure supplement(s) for figure 5:

**Figure supplement 1.** Structure determination of apamin-bound SK2–4/CaM and compound 1-bound SK2–4/CaM.

apamin prior to cryo-EM grid preparation. 3D reconstruction in C4 symmetry produced a map with additional density at the extracellular opening of the K⁺ pore consistent with the size of a single apamin molecule bound per tetramer. However, the C4 symmetric density corresponds to asymmetric apamin bound to four distinct but structurally equivalent orientations, which prevented apamin fitting. To more accurately align particles and improve the apamin density, we used a combination of focused classification and local refinements to generate a reconstruction in C1 symmetry to a final resolution of 3.2 Å (*Figure 2—figure supplement 2C*, *Figure 5—figure supplement 1*, *Supplementary file 1*). Elongated density in the C1 map at the extracellular opening of the pore allowed for accurate

placement of the C-terminal helix (residues 6–17) from a nuclear magnetic resonance (NMR) structure of apamin (*Figure 5A*; *Kuzmenkov et al., 2022*). Two well-defined tubular densities at the center of the helix were confidently modeled with the side chains of Arg13 and Arg14.

Overall, the apamin-bound SK2–4/CaM structure resembles $Ca^{2+}$-bound SK2–4. The N-terminal lobe of CaM engages with the $S_{45}A$ helix, the S5 and S6 helices adopt a similar conformation, and the intracellular gate, Val390, is open with a radius of 3.5 Å (*Figure 2D*). The most significant conformational change is in the position of the S3–S4 linker, which shifts ~2 Å away from the pore axis to accommodate apamin binding (*Figure 5C*). The C-terminal helix of apamin binds at the extracellular opening of the pore and interacts with all four S3–S4 linkers of the tetramer (*Figure 5A*). Essential residues Arg13 and Arg14 insert into the Phe243 constriction to form cation-π interactions (*Figure 5A, C*; *Vincent et al., 1975*). In this position, apamin blocks the exit of $K^+$ ions from the extracellular side of the pore to inhibit conduction.

To probe the interaction between SK2 Phe243 and apamin Arg13 and Arg14, we determined the apamin sensitivity of a Phe243Ala mutant and found it to be insensitive to apamin inhibition up to 3 µM (*Figure 5B*), demonstrating the critical role of this interaction for apamin binding and inhibition. Prior mutagenesis of both the S3–S4 linker and the outer pore regions further supports the observed apamin-binding site and proposed mechanism of apamin inhibition. For example, mutation of Ser244 and Tyr245 in the S3–S4 linker reduces the potency of apamin by 10- and 5-fold, respectively (*Nolting et al., 2007*; *Weatherall et al., 2011*). Ser244 is within 5 Å of the apamin-binding site and bulky mutations may clash with apamin (*Figure 5A*). Tyr245 hydrogen bonds with Asp363, likely stabilizing the S3–S4 linker conformation for apamin binding (*Figure 3C*). In the outer pore, mutation of His336 decreases sensitivity of SK2 to apamin (*Lamy et al., 2010*). As discussed above, this residue also stabilizes the S3–S4 linker conformation (*Figure 3B*). Notably, the previous mutagenesis experiments decreased but did not fully abolish the potency of apamin as we observed for the Phe243Ala mutation, underscoring that the interaction between apamin and Phe243 is essential.

Fortuitously, the apamin-bound structure provides some insight into S3–S4 linker dynamics and the potential transition from a closed to open extracellular constriction. Insertion of apamin Arg13 and Arg14 dilates the Phe243 constriction from 1.8 to 3 Å (*Figures 2D and 5C*). Such an S3–S4 linker conformation in the absence of apamin would permit $K^+$ ion conduction, providing an example of the type of S3–S4 movement required to expand the extracellular constriction. Compared with the apamin-free $Ca^{2+}$-bound conformation (*Figure 4A*), the movement of the S3–S4 linker away from the pore axis upon apamin binding is accompanied by multiple conformational changes in the selectivity filter to adopt the canonical $K^+$-selective conformation with four $K^+$ coordination sites: Arg240 withdraws and frees Asp363 to form a weak hydrogen bond with a rotated Trp350 side chain while Tyr361 in the selectivity filter is rotated 180° and forms a hydrogen bond with Thr355 in the pore helix (*Figure 4—figure supplement 1C*, *Figure 5C, D*). Toxin-induced selectivity filter conformational change has also been reported for $K_v1.3$ with the sea anemone toxin ShK. However, unlike apamin binding to SK2–4, ShK binds directly to the $K_v1.3$ selectivity filter to convert a C-type inactivated conformation to a canonical $K^+$ selective structure with four coordination sites (*Tyagi et al., 2022*; *Selvakumar et al., 2022*). The change in selectivity filter conformation in apamin-bound SK2–4 seems to be driven instead by the weakening of interactions between the selectivity filter and the S3–S4 linker. Based on the structure of SK2–4 bound to apamin, we hypothesize that in the physiological conductive/dilated state of the extracellular constriction, the S3–S4 linker and Phe243 will withdraw from the pore axis and the selectivity filter will return to the canonical conformation with 4 $K^+$ coordination sites to maintain the $K^+$ selectivity demonstrated for SK2 (*Köhler et al., 1996*; *Desai et al., 2000*; *Park, 1994*; *Jäger et al., 2000*).

## Identification and functional characterization of novel SK2 modulators

To identify new modulators of SK2, we developed a high-throughput patch clamp assay using CHO cells stably expressing WT human SK2 channels on the Qube instrument (Sophion) (see methods section for details). Inhibition or activation of SK channels by compounds was characterized in the presence of 2 µM intracellular free $Ca^{2+}$. Small molecule library screening and subsequent rounds of medicinal chemistry optimization produced compound 1 as a potent inhibitor of SK2 with an $IC_{50}$ of 69 nM (*Figure 6A, B*). Selectivity profiling demonstrated that compound 1 has a 10-fold reduced potency for SK4 ($IC_{50}$ of 660 nM) but is selective over other ion channels tested including hERG,

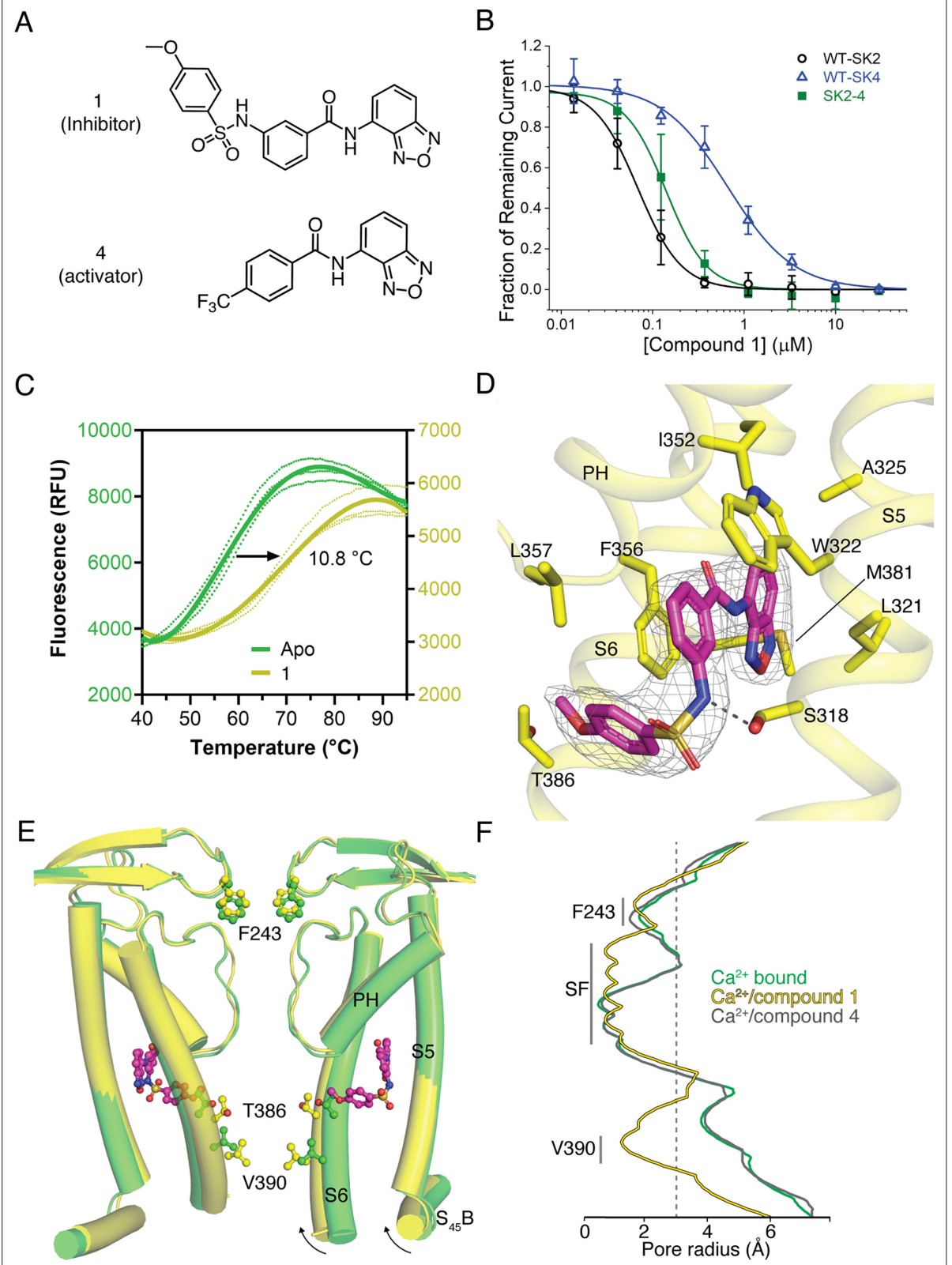

**Figure 6.** Mechanism of compound 1 inhibition. (**A**) Structure of compound 1 (inhibitor) and compound 4 (activator). (**B**) Potency of compound 1 on SK2 (black circles, IC$_{50}$ = 69 nM), SK2–4 (green squares, IC$_{50}$ = 140 nM), and SK4 (blue triangles, IC$_{50}$ = 660 nM). Change in SK current amplitude for different cell populations each exposed to a different extracellular compound 1 concentration was normalized to DMSO control to construct dose responses. Data points represent mean ± SD ($n$ = 4). (**C**) Melting curves of Ca$^{2+}$-bound SK2–4 in the absence (green, $T_m$ = 58.7°C) and presence (yellow, $T_m$ = 69.5°C)

*Figure 6 continued on next page*

*Figure 6 continued*

of compound 1 measured by CPM indicates target engagement. Dotted lines show individual replicates while solid lines show the nonlinear fit of all data for each condition. Raw data were analyzed by fitting melt curves using an extended Boltzmann function to determine melting temperature $T_m$ for each replicate, and then taking the mean of the three replicates for each condition. Thermal stabilization value was determined by subtracting the mean $T_m$ value of the apo samples from the mean $T_m$ value of the compound 1 samples. (**D**) Compound 1 (magenta sticks, density shown at 5 σ) interacts with a pocket formed by the S5, pore helix, and S6. Ser318 of S5 is in position to hydrogen bond (dashed line) with the sulfonamide nitrogen of compound 1. (**E**) Overlay of the $K^+$ pore of $Ca^{2+}$-bound SK2–4 (green) and compound 1-bound SK2–4 (yellow) (only 2 subunits are shown for clarity). The methoxy of compound 1 (magenta spheres) clashes with S6 Thr386 (spheres) and induces a movement of the S6 and $S_{45}B$ helices toward the pore axis (arrow) to close the intracellular gate (Val390, spheres). (**D**) Pore radii of $Ca^{2+}$-bound (green), compound 1-bound (yellow), and compound 4-bound (gray) SK2–4. The location of the intracellular gate (Val390), selectivity filter (SF), and extracellular constriction (Phe243) is indicated. Gray dashed line indicates the radius of hydrated $K^+$ (3 Å).

The online version of this article includes the following figure supplement(s) for figure 6:

**Figure supplement 1.** Characterization of compound 1 inhibition.

**Figure supplement 2.** Nuclear magnetic resonance (NMR) spectra for compound 1.

Nav1.5, KCNQ1, and Cav1.2 (*Figure 6B*, *Figure 6—figure supplement 1A*). Compound 1 inhibits the SK2–4 chimera with similar potency as WT SK2 (*Figure 6B*), suggesting binding at the transmembrane or extracellular domains. Furthermore, we used differential scanning fluorimetry (DSF) to confirm binding of compound 1 to SK2–4. Addition of compound 1 resulted in a strong thermal stabilization of 10.8°C, indicative of target engagement (*Figure 6C*).

To characterize the mechanism of inhibition, we determined the co-structure of compound 1 bound to SK2–4/CaM in the presence of $Ca^{2+}$ at 3.3 Å resolution (*Figure 2—figure supplement 2D*, *Figure 5—figure supplement 1*, *Supplementary file 1*). Four molecules of compound 1, which is clearly defined by cryo-EM density, were bound per SK2–4 tetramer at the interface of the S5, pore helix, and S6 of each subunit (*Figure 6D, E*). The benzoxadiazole moiety of compound 1 rests in a hydrophobic pocket lined by Leu321 and Ala325 of S5, Ile352 and Phe356 of the pore helix, and Met381 of S6. In the $Ca^{2+}$-bound and $Ca^{2+}$-free apo conformations of SK2–4/CaM, this pocket is blocked by Leu321 adopting an alternative rotamer conformation (*Figure 6—figure supplement 1B, C*). The central benzamide moiety is positioned between Trp322 of S5 and Phe356 and Leu357 at the C-terminus of the pore helix. The sulfonamide tail of compound 1 extends toward but does not enter the ion pore, and the sulfonamide nitrogen hydrogen bonds with Ser318 on S5 (*Figure 6D, E*). The methoxyphenyl forms an edge-to-face interaction with Phe356, and the methoxy contacts Thr386 on S6, which is one N-terminal helical turn removed from the intracellular gate at Val390 (*Figure 6E*). The residues that interact with compound 1 are conserved in SK1–3, and thus compound 1 is likely not selective among these isoforms (*Figure 1—figure supplement 1*). However, for SK4, Thr212 replaces SK2 Ser318, and Trp216 (homologous to SK2 Trp322) is conserved but adopts a different rotamer conformation (*Figure 6—figure supplement 1D*). Both changes occlude the compound 1-binding site in SK4 and would likely reduce compound 1 potency on SK4 as observed in the functional data.

Compared with the ligand-free $Ca^{2+}$-bound conformation, there are two significant conformational changes in the structure of SK2–4/CaM bound to compound 1. First, there are slight changes in the position of the S5 and pore helix to accommodate the compound 1 core (*Figure 6E*). These movements translate through Tyr335 and His336 at the extracellular end of S5 to the S3–S4 linker, which is shifted by 1.2 Å in the extracellular direction (*Figures 3B and 6E*, *Figure 6—figure supplement 1E, F*). In this conformation, the hydrophobic constriction at Phe243 would block ion conduction (*Figure 6F*); however, non-continuous side chain density and a high b-factor (51 $Å^2$ compared to 33 $Å^2$ in the $Ca^{2+}$-bound state) indicate increased Phe243 flexibility (*Figure 4—figure supplement 1D*). Movement of the S3–S4 linker coincides with conformational changes in the selectivity filter similar to those observed with apamin-bound SK2–4 (described above) resulting in a selectivity filter with four $K^+$ coordination sites (*Figure 2—figure supplement 2D*, *Figure 4—figure supplement 1D*, *Figure 6—figure supplement 1E, F*). This structure further supports the hypothesis that movement of the S3–S4 linker away from the selectivity filter induces a conformation with four $K^+$ coordination sites.

The second notable conformational change is in the position of the S6 helices. Structural comparisons demonstrate that the methoxy group of compound 1 would clash with Thr386 on S6 in the $Ca^{2+}$-bound apo conformation of SK2–4 (*Figure 6E*, *Figure 6—figure supplement 1B*). To accommodate the methoxy, Thr386 and the portion of the S6 helices C-terminal to Thr386 shift toward

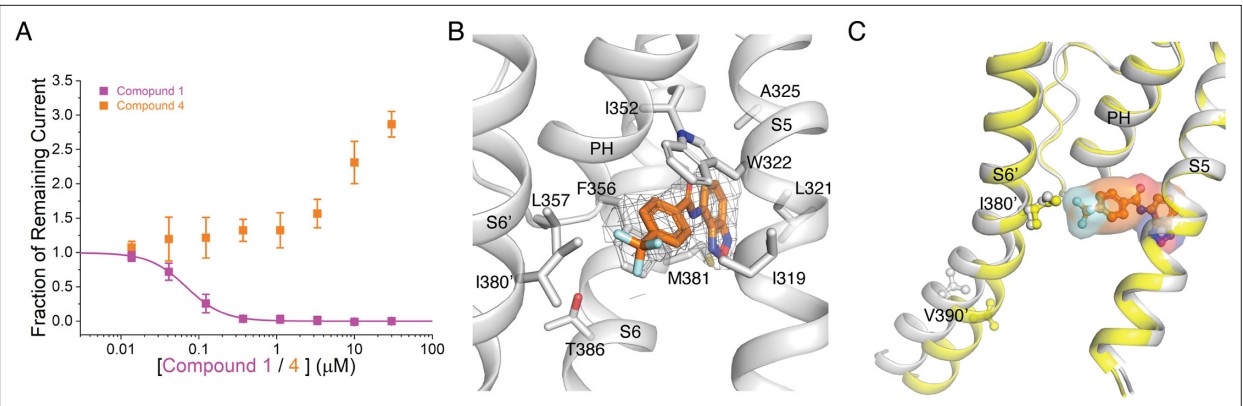

**Figure 7.** Mechanism of compound 4 activation. (**A**) Activation or inhibition of SK2 channel by compound 4 (orange squares) and compound 1 (magenta squares), respectively. Change in WT hSK2 current amplitude for different cell populations each exposed to a different extracellular compound concentration was normalized to DMSO control to construct dose responses. Intracellular solution contains 2 μM free $Ca^{2+}$. SK current was isolated by subtracting leak current determined at the end of experiment by adding saturating concentration of UCL1684, a pore blocker whose potency is not affected by compound 1 or 4. Data points represent mean ± SD (*n* = 6). (**B**) Compound 4 (orange sticks, density shown at 5 σ) interacts with a pocket formed by the S5, pore helix, and S6 and the trifluoromethyl extends toward Ile380 on the neighboring S6 (**S6'**). (**C**) Overlay of compound 1-bound SK2–4 (yellow) and compound 4-bound (gray) SK2–4. In the closed state of the S6 helices (yellow cartoon), the trifluoromethyl of compound 4 clashes with Ile380 (yellow sticks) on the S6 helix (**S6'**) of the neighboring subunit (2.3 Å distance). In the open state of the S6 helices (gray cartoon), Ile380 (gray sticks) is 3.1 Å distant from the trifluoromethyl minimizing this clash.

The online version of this article includes the following figure supplement(s) for figure 7:

**Figure supplement 1.** Structure determination of compound 4-bound SK2–4/CaM.

**Figure supplement 2.** Nuclear magnetic resonance (NMR) spectra for compound 4.

the pore axis akin to the S6 helix shift observed in the $Ca^{2+}$-free conformation (*Figure 6—figure supplement 1G*). Concomitantly, the pore radius at the Val390 intracellular gate is reduced to 1.2 Å demonstrating a clear mechanism for channel inhibition (*Figure 6F*). Notably, compound 1 closes the gate at Val390 in the presence of saturating $Ca^{2+}$ concentrations, demonstrating that this mechanism of inhibition can override the $Ca^{2+}$-dependent gating. The importance of the methoxy for inhibition was demonstrated by structurally related compounds 2 and 3, in which the benzoxadiazole of compound 1 is replaced with a benzothiadiazole isostere (*Figure 6—figure supplement 1H, I*). Removal of the methoxy from compound 2 produced compound 3 with a 30-fold reduction in potency.

A previous report identified a pair of structurally related compounds that act as selective SK1 inhibitors and activators. This chemical series requires Ser293 on S5 for activity, which corresponds to Leu321 in SK2, suggesting that the SK1 modulators and compound 1 share the same binding site (*Figure 1—figure supplement 1*; *Hougaard et al., 2012*). We hypothesized that compound 1 analogs lacking the methoxyphenyl, which our structure and functional data demonstrated is important for SK2 inhibition, may activate SK2. Indeed, a focused screen of compound 1 analogs lacking the methoxyphenyl identified compound 4 as an SK2 activator (*Figures 6A and 7A*) which dose-dependently increases SK2 current in the presence of 2 μM intracellular $Ca^{2+}$. Like compound 1, compound 4 retains the benzoxadiazole/benzamide core, but the benzamide phenyl features a trifluoromethyl substituent para to the amide. In the 3.1 Å structure of $Ca^{2+}$-bound SK2–4 in complex with compound 4, clear cryo-EM density demonstrates that the benzoxadiazole/benzamide core occupies the same pocket as observed for compound 1 (*Figure 7B*, *Figure 7—figure supplement 1*, *Supplementary file 1*). However, the S6 helices are in an open conformation, and the trifluoromethyl interacts with Ile380, which is 2.5 helical turns N-terminal to Val390, on the S6 helix from a neighboring subunit (*Figures 6F and 7B*). Structural overlays predict that the trifluoromethyl would clash with Ile380 in the closed conformation of the S6 helices and thereby promote the open conformation to activate SK2 (*Figure 7C*). Unlike compound 1, compound 4 binding does not induce movement of the S3–S4 linker and the selectivity filter retains a conformation with two $K^+$ coordination sites (*Figure 6F*, *Figure 7—figure supplement 1G*).

## Discussion

The structures of an SK2–4 chimeric channel in the $Ca^{2+}$-bound and $Ca^{2+}$-free conformations revealed two unexpected features of the SK2 transmembrane domains. First, the S3–S4 linker forms an anti-parallel β-turn that extends over the S5–S6 segments and interacts with the extracellular pore loops (*Figure 3*). In this conformation, Phe243 on the S3–S4 linker is positioned directly above the selectivity filter creating a hydrophobic constriction with a radius of 1.8 Å, which prevents the efflux of hydrated $K^+$ ions (*Figures 2D, 3A, C, and 4A*). Therefore, the $Ca^{2+}$-bound SK2–4 structure represents a non-conductive conformation even though the intracellular gate is open. Second, the selectivity filter adopts a conformation that resembles non-selective cation channels, such as NaK and HCN, with only two $K^+$ coordination sites (*Shi et al., 2006*; *Lee and MacKinnon, 2017*; *Figure 4A, E, F*, *Figure 4—figure supplement 1*). This selectivity filter conformation seems to be due to the interaction between Asp363 at the C-terminus of the selectivity filter and residues Arg240 and Tyr245 on the S3–S4 linker. In the related SK4 channel, the homologous aspartate hydrogen bonds with a pore helix tryptophan (*Lee and MacKinnon, 2018*; *Figure 4C*), which is critical for the formation of a selectivity filter with four $K^+$ coordination sites observed in all $K^+$ selective channels to date (*Derebe et al., 2011*; *Sauer et al., 2011*). Therefore, by sequestering Asp363 and preventing its interaction with Trp350 in the pore helix, the S3–S4 linker likely induces the two $K^+$ coordination site selectivity filter observed in the apo SK2–4 structures (*Figure 4A*, *Figure 4—figure supplement 1A, B*). In support of this hypothesis, a similar S3–S4 linker and selectivity filter conformation was observed in the structure of rat SK2, which was published during the preparation of this manuscript, and mutagenesis demonstrated that the interactions between the S3–S4 linker and pore residues induce the observed selectivity filter with two $K^+$ coordination sites (*Nam et al., 2025*). However, this selectivity filter conformation seems to only exist in a non-conductive state of the channel due to the extracellular constriction at Phe243 and thus should not affect the $K^+$ selectivity observed for SK2 and SK2–4 (*Figure 1B*; *Köhler et al., 1996*; *Desai et al., 2000*; *Shi et al., 2006*; *Derebe et al., 2011*; *Sauer et al., 2011*; *Lee and MacKinnon, 2017*; *Park, 1994*; *Jäger et al., 2000*).

The subsequent structures of SK2–4 bound to apamin and compound 1 elucidated how the SK channels maintain $K^+$ selectivity in the conductive state. Conductivity likely requires movement of the S3–S4 linker and Phe243 away from the pore axis to expand the extracellular constriction. Such a shift was observed upon binding of both apamin and compound 1 and was accompanied by four structural rearrangements: (1) Asp363 no longer hydrogen bonds with Arg240 in the S3–S4 linker, (2) Asp363 reorients toward Trp350 in the pore helix, (3) Trp350 rotates to enable a hydrogen bond with Asp363, and (4) Tyr361 in the selectivity filter rotates 180° and forms a hydrogen bond with Thr355 in the pore helix (*Figure 4—figure supplement 1C, D*, *Figure 5C, D*, *Figure 6—figure supplement 1E, F*). These structural changes produce a canonical $K^+$-selective selectivity filter with four $K^+$ coordination sites. Based on these structures, we predict that in a conductive state of SK2, the extracellular constriction dilates to allow for the flow of $K^+$ ions. Such a movement would weaken the interactions between the S3–S4 linker and selectivity filter, producing a selectivity filter conformation with four $K^+$ coordination sites to maintain the $K^+$ selectivity.

The physiological role of the S3–S4 linker and the mechanism of extracellular constriction opening require further investigation. One possibility is that the S3–S4 linker and extracellular constriction exists in an equilibrium between conductive and non-conductive conformations. Such a mechanism may explain some properties of SK2 that are not observed in SK4, which lacks a structured S3–S4 linker, such as its low conductance (~10 pS) and the ability to switch between low- and high-open probability states (*Köhler et al., 1996*; *Hirschberg et al., 1998*). Indeed, mutation of Phe243 in rat SK2 produced a twofold increase in channel conductance (*Nam et al., 2025*). Alternatively, other physiological factors, such as PIP2 (*Woltz et al., 2024*; *Zhang et al., 2014*) or protein–protein interactions (*Lin et al., 2008*; *Luján et al., 2018*; *García-Negredo et al., 2014*), may exist in live cells that modulate the interaction between S3–S4 linker and the selectivity filter. Importantly, the extracellular constriction provides novel opportunities for modulation of SK2.

The structures and functional experiments presented here revealed two binding sites for SK2 modulators and identified three distinct mechanisms of modulation. The bee toxin apamin binds at the extracellular opening of the pore, at the same site as UCL1684, to a site co-formed by the S3–S4 linker from each subunit, explaining why both the S3–S4 linker and pore residues were found to be important for apamin binding and inhibition (*Figure 5*; *Nolting et al., 2007*; *Weatherall et al., 2011*;

*Lamy et al., 2010*; *Nam et al., 2025*). The two essential arginine residues (13 and 14) of apamin are directed toward the selectivity filter and form cation–π interactions with the essential S3–S4 linker residue Phe243, blocking $K^+$ ion efflux and inhibiting conduction. In support of this binding interaction and mechanism of inhibition, a Phe243Ala mutant abolishes apamin inhibition (*Figure 5B*). The elucidated apamin-binding site and mechanism of inhibition may enable development of new modulators that target the extracellular domains of SK2.

A second partially cryptic binding pocket at the interface of the S5, pore helix, and S6 was revealed through the identification and characterization of compounds 1 and 4 (*Figures 6D and 7B*). This binding pocket interacts with a benzoxadiazole/benzamide core and the functional arm on the benzamide phenyl interacts with the S6 helices to modulate SK2 activity. For compound 1, a sulfonamide methoxyphenyl meta to the amide contacts Thr386 on the S6 helix in the same subunit to induce the closed conformation of the S6 helices and inhibit the channel (*Figure 6D–F*). Conversely, for compound 4, the trifluoromethyl para to the amide contacts Ile380 on the S6 helix from a neighboring subunit to promote the open conformation of the S6 helices and activate the channel (*Figures 6F and 7B, C*). This proposed mechanism of modulation suggests that compound 1 may bind preferentially to the closed conformation of the S6 helices, and compound 4 may bind preferentially to the open conformation of the S6 helices. Further optimization of the functional groups on the benzamide phenyl that interact with S6 helices may produce more potent activators and/or inhibitors. For example, more potent activation may be achieved if the interaction between compound 4 and the neighboring S6 was closer to the intracellular gate at Val390 (i.e., within 1 helical turn). Producing isoform-selective modulators targeting the compound 1/4-binding pocket may prove more challenging due to the high sequence conservation of residues lining the pocket among SK1–3. However, SK1 modulators that are predicted to share the same binding site as compound 1 take advantage of a unique S5 Ser to confer selectivity over SK2 and SK3, suggesting isoform selectivity could be achieved through further optimization (*Hougaard et al., 2012*).

In summary, this study characterized SK2 channel dynamics as well as mechanisms of pharmacological modulation utilizing a SK2–4 chimera, containing the S1–S6 transmembrane regions of human SK2 and the intracellular domains of human SK4. SK2–4 is $K^+$-selective, activated by $Ca^{2+}$, sensitive to SK2 modulators that bind the S1–S6 transmembrane regions, and suitable for cryo-EM structure determination. The findings revealed critical structural features and mechanisms of SK2 channel modulation, providing a framework for the development of targeted therapeutics for the SK channel family.

## Methods

### Construct design

The amino acid sequences for human SK2 (KCNN2, Uniprot Q9H2S1, 579 amino acids) and human SK4 (KCNN4 Uniprot O15554, 427 amino acids) were used to design the chimeric SK2–4 channel construct. The N-terminal portion of SK2 made up of residues 1–123 was replaced with SK4 residues 1–15. The C-terminal portion of SK2 encompassing residues 413–579 was replaced with SK4 residues 306–428. An HRV3C protease recognition sequence followed by a GFP sequence was appended at the C-terminus of SK2–4 resulting in the SK2–4-GFP construct.

The amino acid sequences for SK2–4-GFP and human CaM (CALM1, Uniprot P0DP23, 149 amino acids) were codon optimized using Twist Bioscience's tool to create DNA sequences for expression. These constructs were synthesized and cloned into the vector pWIL-BacMam for expression and purification. The pWIL-BacMam vector contains the Tn7 transposon and encodes a baculovirus genome with a multiple cloning site, such that when a gene of interest is placed in the pWIL-BacMam vector and transformed into *E. coli* DH10Bac competent cells (Thermo Fisher Scientific), it produces bacmid containing the gene of interest. It also contains a CMV enhancer and promoter for expression of the gene of interest in mammalian cells.

### SK2–4/CaM chimeric channel complex production

SK2–4 chimeric channel/CaM complex was produced using the BacMam method. Plasmids encoding the constructs (SK2–4-GFP, CaM) were each separately transformed into *E. coli* DH10Bac competent cells (Thermo Fisher Scientific) following the manufacturer's protocol and plated onto blue/white selection LB agar plates (Teknova). After 2 days of incubation at 37°C, a white colony for

each construct was selected and grown overnight in 5 ml liquid medium containing 7 μg/ml genta-micin sulfate, 50 μg/ml kanamycin, and 10 μg/ml tetracycline hydrochloride (Teknova). Cells were harvested by centrifugation at 5000 × $g$ for 5 min. Cell pellets were resuspended in 250 μl P1 buffer (QIAGEN) and incubated briefly with 250 μl P2 buffer (QIAGEN) before addition of 350 μl N3 buffer (QIAGEN). Solutions were centrifuged for 10 min at 16,000 × $g$. Supernatant was added to 1 ml of ice-cold isopropanol and incubated for 20 min at –20°C. Solutions were centrifuged for 15 min at 16,000 × $g$ and 4°C to pellet DNA. Supernatant was discarded and pellet was washed with 1 ml of ice-cold 70% ethanol, followed by centrifugation at 16,000 × $g$ for 10 min. Supernatant was discarded, and the pellet was left to air-dry for 10 min before resuspension with 50 μl of molecular-biology grade water.

Baculovirus for each construct was produced by transfection of bacmid into SF9 cells as follows: 3 μl X-tremeGENE HP DNA transfection reagent (Roche), 5 μl of bacmid, and 100 μl transfection medium (Expression Systems) were incubated for 15 min at room temperature. After incubation, 2.5 ml of Sf9 insect cells (Expression Systems) at a density of 1 million cells/ml were added. Cells were incubated for 1 week with shaking at 27°C before baculovirus-containing medium was harvested. Baculovirus was amplified in Sf9 cells for two additional rounds after transfection. Expi293F cells (Thermo Fisher Scientific) were grown in suspension at 37°C in 8% $CO_2$ atmosphere with 110 rpm shaking to a density of approximately 3 × $10^6$ cells/ml before transduction with virus. A virus was added to the culture with 2% vol/vol CaM baculovirus and 8% vol/vol of SK2–4-GFP baculovirus. Approximately 16 hr after transduction, sodium butyrate was added to cultures to a final concentration of 10 mM. Cultures were incubated an additional ~48 hr at 37°C in 8% $CO_2$ atmosphere with 110 rpm shaking before harvest by centrifugation at 6000 × $g$ for 45 min. Cell pellets were flash-frozen in liquid nitrogen and stored at –80°C until use.

Cell pellet from a 4-l culture was resuspended in 200 ml of room-temperature resuspension buffer (10 mM Tris pH 8, 20 mM KCl, 2 mM $CaCl_2$, 0.5 mM $MgCl_2$, 0.05 mg/ml DNase I, Pierce protease inhib-itor tablet EDTA-free) for 30 min with vigorous stirring. Lysate was centrifuged for 45 min at 35,500 × $g$ to collect membranes. The membrane pellet was resuspended in 200 ml of extraction buffer (10 mM Tris pH 8, 20 mM KCl, 2 mM $CaCl_2$, 2% wt/vol $n$-dodecyl-β-D-maltopyranoside (DDM), 0.4% wt/vol cholesteryl hemisuccinate tris salt (CHS), Pierce protease inhibitor tablet EDTA-free) using a dounce. Membranes were solubilized with vigorous stirring at 4°C for 2 hr. The solution was clarified by ultra-centrifugation at 195,000 × $g$ for 1 hr at 4°C.

The supernatant was incubated with 4 ml GFP nanobody resin (Bulldog Bio) pre-equilibrated with wash buffer (20 mM Tris pH 8, 150 mM KCl, 2 mM $CaCl_2$, 0.05% wt/vol DDM, 0.01% wt/vol CHS) for 1.5 hr at 4°C with gentle agitation. Resin was collected over a gravity column and washed with 5 column volumes (CV) of wash buffer supplemented with 5 mM ATP and 10 mM $MgCl_2$, followed by washing with 10 CV of unsupplemented wash buffer. Resin was resuspended in 20 ml of wash buffer with 0.25 mg of HRV3C protease and incubated overnight at 4°C with gentle agitation to cleave untagged SK2–4/CaM complex from the resin. Supernatant containing protein was collected and concentrated to 0.5 ml using 100 kDa MWCO Amicon Ultra centrifugal filter (EMD Millipore).

Concentrated protein was applied to a Superose 6 Increase 10/300 GL column (Cytiva) pre-equilibrated with SEC buffer. For samples purified in the presence of $Ca^{2+}$, the SEC buffer was 20 mM Tris pH 8, 150 mM KCl, 2 mM $CaCl_2$, 0.005% wt/vol glyco-diosgenin (GDN), 0.0005% wt/vol CHS. For samples purified in the absence of $Ca^{2+}$, the SEC buffer was 20 mM Tris pH 8, 150 mM KCl, 5 mM EGTA, 0.005% wt/vol GDN, 0.0005% wt/vol CHS. Size exclusion was performed at 4°C with a flow rate of 0.5 ml/min. SDS–PAGE was performed using NuPAGE Bis-tris 4–12% gel (Invitrogen) and NuPAGE MES SDS running buffer (Invitrogen) to identify fractions with pure SK4/CaM complex. Fractions were collected and concentrated using 100 kDa MWCO Amicon Ultra centrifugal filter to desired concen-tration and frozen directly onto cryo-EM grids.

## EM sample preparation and data collection

SK2–4/CaM complex samples were prepared at the following concentrations: $Ca^{2+}$-bound SK2–4/CaM at 7.5 mg/ml; $Ca^{2+}$-free SK2–4/CaM at 8.25 mg/ml. For samples with apamin present, $Ca^{2+}$-bound SK2–4/CaM was supplemented with apamin (Sigma-Aldrich) to a final concentration of 200 μM. For the sample with compound 1, $Ca^{2+}$-bound SK2–4/CaM was supplemented with compound 1 in 100% DMSO to a final concentration of 200 μM compound 1 and 2% DMSO. For the sample with compound

4, $Ca^{2+}$-bound SK2–4/CaM was supplemented with compound 4 in 100% DMSO to a final concentration of 500 μM compound 4 and 2% DMSO.

UltrAuFoil R 0.4/1.2 200 mesh grids were used for the $Ca^{2+}$-bound and $Ca^{2+}$-free SK2–4/CaM samples and UltrAuFoil R 1.2/1.3 300 mesh grids were used for all other samples. The grids (Electron Microscopy Sciences) were glow-discharged for 30 s at 0.5 mBar with 15 mA using the easiGlow system (PELCO). 5 μl of sample was applied to grids at 4°C with 100% humidity. After 30 s, grids were blotted for 5 s with blot force 25 and plunged into liquid ethane using the Vitrobot Mark IV system (Thermo Fisher).

For $Ca^{2+}$-bound SK2–4/CaM, $Ca^{2+}$-free SK2–4/CaM, SK2–4/CaM + Apamin, and $Ca^{2+}$-bound SK2–4/CaM + compound 1, data were collected on a Titan Krios microscope (Thermo Fisher Scientific, Waltham, MA, USA) equipped with a Cs-corrector hardware operated at an accelerating voltage of 300 kV with a 50 μm C2 aperture at an indicated magnification of 75,000× in nanoprobe mode. A Falcon4i Direct Electron Detector camera operated in Electron-Event representation mode was used to acquire dose-fractionated images with Thermo Fisher Scientific EPU software. For $Ca^{2+}$-bound SK2–4/CaM + compound 4, data were collected on a Glacios microscope (Thermo Fisher Scientific, Waltham, MA, USA) operated at an accelerating voltage of 200 kV with a 50 μm C2 aperture at an indicated magnification of ×120,000 in nanoprobe mode. A Falcon3 Direct Electron Detector camera was used to acquire dose-fractionated images with Thermo Fisher Scientific EPU software. Data collection parameters for each data set are listed in *Supplementary file 1*.

## EM data processing

[$Ca^{2+}$-bound SK2–4/CaM] Movie stacks were gain-corrected and motion-corrected in patches using MotionCor2 (*Zheng et al., 2017*). Motion-corrected images were imported into cryoSPARC4 (*Punjani et al., 2017*). Contrast transfer function parameters were determined using Patch CTF. Blob picking was performed with a subset of the micrographs with an expected diameter of 160–180 Å. Selected particles were extracted with a box size of 360 pixels and subjected to reference-free 2D classification to generate templates, which were used for subsequent template-based autopicking with the full set of micrographs. 8,310,008 particles were selected and extracted with a box size of 360 pixels. After iterative rounds of reference-free 2D classification, 910,204 particles were used to generate two ab initio models. The 658,368 particles from the better model were subjected to non-uniform 3D refinement (*Punjani et al., 2020*) with imposed C4 symmetry using the ab initio model as a template resulting in a 3.1 Å consensus refinement. After local CTF refinement, a soft mask excluding the detergent micelle was generated for local refinement with imposed C4 symmetry, resulting in the final consensus refinement with a resolution of 3.1 Å.

[$Ca^{2+}$-free SK2–4/CaM] Movie stacks were gain- and motion-corrected in patches using MotionCor2 (*Zheng et al., 2017*). Motion-corrected images were imported into cryoSPARC4 (*Punjani et al., 2017*). Contrast transfer function parameters were determined using Patch CTF. Blob picking was performed with a subset of the micrographs with an expected diameter of 160–180 Å. Selected particles were extracted with a box size of 360 pixels and subjected to reference-free 2D classification to generate templates, which were used for subsequent template-based autopicking with the full set of micrographs. 8,160,096 particles were selected and extracted with a box size of 360 pixels. After iterative rounds of reference-free 2D classification, 386,845 particles were subjected to non-uniform 3D refinement (*Punjani et al., 2020*) with imposed C4 symmetry using the $Ca^{2+}$-bound SK2–4/CaM consensus refinement as a template, resulting in a 3.4-Å consensus refinement. A soft mask excluding the detergent micelle and the intracellular stalk was generated for local refinement with imposed C4 symmetry, resulting in the final consensus refinement with a resolution of 3.4 Å.

[SK2–4/CaM + apamin] Movie stacks were gain- and motion-corrected in patches using MotionCor2 (*Zheng et al., 2017*). Motion-corrected images were imported into cryoSPARC4 (*Punjani et al., 2017*). Contrast transfer function parameters were determined using Patch CTF. Template autopicking was performed using the 2D class averages from the SK2–4/CaM + $Ca^{2+}$ dataset. 7,528,863 picked particles were extracted using a box size of 360 pixels. Iterative rounds of reference-free 2D classification led to 1,440,270 particles which were used to generate two ab initio models. The better model comprising 1,186,640 particles was subjected to non-uniform refinement (*Punjani et al., 2020*) using the ab initio model as a template, first with C4 symmetry imposed, which led to a consensus refinement at 3 Å. Density at the top of the SK2–4 pore was

identified as the likely apamin–binding site. A soft ovoid mask covering the top of the pore and the SK2–4 selectivity filter was used for focused 3D classification without alignment of the particles with C1 symmetry and five classes. Three classes comprising 734,192 total particles had clear asymmetrical density for apamin at the extracellular opening of the pore. The classes appeared to be structurally identical but rotated by 90° from one another. Volume alignment tools were used to rotate the volumes and associated particle stacks 90° or 180° around the symmetry axis so the three classes were in the same orientation. These particle stacks were put into a local refinement with C1 symmetry using one of the classes as a template and with a soft mask that excluded the detergent micelle and intracellular stalk of SK2–4, resulting in a consensus refinement at 3.1 Å with clear density for apamin at the extracellular pore opening. This was followed by a non-uniform refinement with C1 symmetry with the local refinement map used as a template, resulting in a final consensus refinement at 3.2 Å.

[$Ca^{2+}$-bound SK2–4/CaM + compound 1] Movie stacks were gain- and motion-corrected in patches using MotionCor2 (*Zheng et al., 2017*). Motion-corrected images were imported into cisTEM (*Grant et al., 2018*) for contrast transfer function estimation followed by particle picking. 1,884,323 selected particles were extracted with a box size of 360 pixels and subsequently imported into cryoSPARC4 (*Punjani et al., 2017*). Reference-free 2D classification was performed to select 326,099 good particles. These were used to generate two ab initio models. The better model comprising 262,274 particles was subjected to non-uniform refinement (*Punjani et al., 2020*) using the $Ca^{2+}$-bound SK2–4/CaM model as a template, resulting in a consensus refinement at 3.3 Å. This refinement was put through global CTF refinement, followed by another round of non-uniform refinement resulting in the final map at 3.3 Å.

[$Ca^{2+}$-bound SK2–4/CaM + compound 4] Movie stacks were gain- and motion-corrected in patches using MotionCor2 (*Zheng et al., 2017*). Motion-corrected images were imported into cryoSPARC4 (*Punjani et al., 2017*). Contrast transfer function parameters were determined using Patch CTF. Template-based autopicking, extraction with a box size of 360 pixels, and reference-free 2D classification were performed to select 436,179 good particles. These were used to generate two ab initio models. The better model comprising 360,727 particles was subjected to non-uniform refinement (*Punjani et al., 2020*) using the $Ca^{2+}$-bound SK2–4/CaM model as a template, resulting in a consensus refinement at 3.1 Å.

## Model building

For the $Ca^{2+}$-bound SK2–4/CaM structure, the SK2–4 region was built de novo in Coot (*Emsley et al., 2010*) using the 3.1 Å -EM density map. For CaM, a chain of CaM from the SK4 structure 6CNN was docked into the cryo-EM map in Chimera (*Pettersen et al., 2004*), followed by adjustment in Coot. Fourfold symmetry was enforced on the model in Coot. The model was refined against the cryo-EM density map using iterative rounds of manual adjustment in Coot followed by real space refinement in Phenix (*Afonine et al., 2018*).

For the model building of the $Ca^{2+}$-free SK2–4/CaM, apamin-bound, compound 1-bound, and compound-4 bound structures, the $Ca^{2+}$-bound SK2–4/CaM structure was used as a starting point and manually docked into the corresponding cryo-EM density maps, followed by manual adjustment in Coot (*Emsley et al., 2010*). For the apamin-bound structure, apamin from the NMR structure 7OXF was docked into the cryo-EM density map using Chimera (*Pettersen et al., 2004*), followed by truncation of unresolved regions and manual adjustment in Coot. For the $Ca^{2+}$-free, compound 1-bound, and compound 4-bound structures, fourfold symmetry was enforced in Coot. No symmetry was enforced for the apamin-bound model. The models were refined against their respective cryo-EM density maps using iterative rounds of manual adjustment in Coot followed by real space refinement in Phenix (*Afonine et al., 2018*).

For all models, if the backbone was not visible in the density map, the residue was not modeled. Residues with poorly defined sidechain density were truncated at Cβ. The quality of the models was determined using MolProbity (*Chen et al., 2010*). Figures were prepared using PyMOL (*Schrödinger, LLC, 2010*). The PyMOL Molecular Graphics System, Version 3.1.3 and Chimera (*Pettersen et al., 2004*).

## Electrophysiology assays

### SK channel assays

All electrophysiology assays for SK used CHO-K1 cells either stably expressing SK channels (WT SK2 and SK4) or transiently transduced with channel baculovirus constructs (WT and mutant SK2, SK4, and SK2–4 chimera) using the BacMam system. Stably channel-expressing cells were cultured using T150 flasks in a 37°C incubator with 5% $CO_2$ until 70–80% confluency. Culture media were composed of Dulbecco's modified Eagle's medium/F-12 nutrient mixture (DMEM/F-12, #31320, Gibco) and 10% fetal bovine serum (#89510-196, Avantor), 1% penicillin/streptomycin (#15140, Gibco) and 1% non-essential amino acids (#11140, Gibco). For BacMam transduction, 8 million parental CHO-K1 cells were seeded into a T150 flask with 15 ml media. After >6 hr in 37°C incubator with 5% $CO_2$ and until cells were attached, media were aspirated and replaced with 10 ml fresh media containing 400 μl baculovirus for individual channel constructs. After this, cells were cultured in a 30°C incubator with 5% $CO_2$. After overnight media were aspirated and replaced with 10 mL media containing 4 μM Tricostatin A (#T-1952, Sigma) for 4 hr before dilution to 1 μM by adding 30 ml media. Cells were ready to use >40 hr in 30°C incubator after transduction. On the day of experiments for all cells, media were aspirated before a wash with PBS (#14190, Gibco). Cells were then dissociated using 5 ml TrypLE (#12605010, Thermo Fisher) for 3 min at 37°C, spun down, and resuspended in serum-free medium (#12052114, Gibco) with added 25 mM HEPES (#15630, Gibco). Right before experiments, cells were spun down again and resuspended in extracellular buffer containing 5 mM $BaCl_2$ at 2–3 million cells/ml before being loaded to the 'cell transfer plate' of the Qube instrument (Sophion).

All SK electrophysiological recordings were made with the automated Qube system and 384X 10 hole QChips at 22°C. Sum of SK currents from up to 10 cells is measured in each recording well from the 384-well plate. Briefly, cells from the Qube 'cell transfer plate' were pipetted into each well to form seals, then broken into whole-cell configuration. Extracellular buffer contains (in mM): 130 NaCl, 6 KCl, 2 $CaCl_2$, 40 sucrose, 10 HEPES, 1 $MgCl_2$, pH = 7.4 with NaOH. To enhance sealing, 5 or 15 mM $BaCl_2$ were added to extracellular buffer (sucrose reduced accordingly to maintain osmolarity) which were washed away with $Ba^{2+}$-free extracellular buffer after the seal was formed. Intracellular buffer contains (in mM): 110 $K_2SO_4$, 8 NaCl, 4.68 $CaCl_2$, 10 HEPES, 5 EGTA, 5 HEDTA, and 4 Mg-ATP added right before experiments, pH = 7.0 with KOH. All SK experiments use this intracellular solution which contains 2 μM free $Ca^{2+}$ (determined by online Maxchelator program available here), except for one case when intracellular free $Ca^{2+}$ was varied to evaluate $Ca^{2+}$-dependent activation as mentioned below. SK current was recorded using a 250ms long voltage ramp from –120 to +80 mV, whereas current level at ~0 mV was taken as the current amplitude to minimize contribution by leak. No series resistance compensation or leak subtraction was included in the voltage protocol. For pharmacological effects, cells were washed with $Ba^{2+}$-free extracellular buffer containing different concentrations of compounds before SK currents were recorded (a single concentration per well). Leak and non-SK currents were estimated at the end of the experiment with saturating concentration of specific SK inhibitors (10 μM UCL1684,30 μM AP14145, or 10 μM TRAM-34, as needed) and subtracted from average current amplitudes. Average current amplitude after compound from each well was normalized to its before-compound amplitude then normalized to time-matched wells treated with 0.3% DMSO. Normalized values at each concentration ($n = 4–6$ wells) were used to construct dose–response curves, which were fitted with Hill Equation to yield $IC_{50}$ values. For $Ca^{2+}$-dependent activation of SK channels, different amounts of $CaCl_2$ were added to intracellular buffer to achieve desired levels of free $Ca^{2+}$ concentration as calculated with Maxchelator. On a 384-well plate, 24 wells (3 columns) were exposed to intracellular buffer containing one of eight different calculated free $Ca^{2+}$ concentrations ranging from 0.1 to 20 μM. Average SK current amplitudes from 24 wells for each $[Ca^{2+}]$ were normalized to the level at 20 μM and plotted against free $Ca^{2+}$ concentration to construct the $Ca^{2+}$-dependent activation curve, which was fitted with the Hill Equation to yield $EC_{50}$ values. Note that on the automated high-throughput electrophysiology system, it is only feasible to record current in whole-cell configuration (in contrast to inside-out) in which true intracellular free $[Ca^{2+}]$ may differ from the calculated values for the bulk intracellular buffers due to incomplete diffusion.

### Counter-screening assays on other ion channels

Compound effect on hERG channel was tested on the Qube system at 35°C using CHO-K1 cells stably expressing hERG channel. Assay conditions were similar to the SK assay except for buffers and voltage

protocol. Extracellular buffer contains (in mM): 145 NaCl, 4 KCl, 2 CaCl$_2$, 10 glucose, 10 HEPES, 1 MgCl$_2$, pH = 7.4 with NaOH. Intracellular buffer contains (in mM): 20 KCl, 120 KF, 10 EGTA, 10 HEPES, pH = 7.2 with KOH. hERG channel inhibition was measured by peak tail current level at –50 mV after a 3.5 s long depolarization to 10 mV. Compound effect on hCav1.2 channel was tested on the Qube system at 35°C using CHO-K1 cells expressing hCav1.2 channel (inducible with the T-Rex system). Assay conditions were similar to the SK assay except for buffers and voltage protocol. Extracellular buffer contains (in mM): 145 NaCl, 10 CaCl$_2$, 10 HEPES, 4 KCl, pH = 7.4 with NaOH. Intracellular buffer contains (in mM): 112 CsCl, 28 CsF, 2 NaCl, 10 HEPES, 8.2 EGTA, pH = 7.3 with CsOH. Cav1.2 channel inhibition was measured by peak current level at 10 mV depolarization from a –70 mV holding potential. The compound effect on the hKCNQ1 channel was tested on the Qube system at 22°C using CHO-K1 cells stably expressing hKCNQ1 channel. Assay conditions were similar to the SK assay except for the voltage protocol. Extracellular buffer contains (in mM): 130 NaCl, 6 KCl, 2 CaCl$_2$, 40 sucrose, 10 HEPES, 1 MgCl$_2$, pH = 7.4 with NaOH. Intracellular buffer contains (in mM) 110 K$_2$SO$_4$, 8 NaCl, 4.68 CaCl$_2$, 10 HEPES, 5 EGTA, 5 HEDTA, and 4 Mg-ATP added right before experiments, pH = 7.0 with KOH. hKCNQ1 channel inhibition was measured by peak current level at 4 s long depolarization to 50 mV from a –80 mV holding potential. The compound effect on the hNav1.5 channel was tested on the QPatch (Sophion) system at room temperature using CHO-K1 cells stably expressing hNav1.5 channel. QPatch is a different automated electrophysiology system than the Qube and uses 48-well plates. Extracellular buffer contains (in mM): 137 NaCl, 4 KCl, 1 MgCl$_2$, 1.8 CaCl$_2$, 10 HEPES, 10 glucose, pH = 7.3 with NaOH. Intracellular buffer contains (in mM): 120 CsF, 20 CsCl, 10 HEPES, 5 EGTA, 10 NaCl, pH = 7.2 with CsOH. hNav1.5 channel inhibition was measured by steady-state peak current level at –10 mV depolarization of 400 ms at 1 Hz from a holding potential of –80 mV. In contrast to Qube assays, compound effect was tested by perfusing individual cells with compounds of increasing concentrations to construct cumulative dose–response curves. All dose–response data points were normalized to time-matched wells/cells treated with DMSO. Dose–response curves were plotted as mean ± SD and fitted with the Hill equation.

## DSF assay

DSF assay was performed in DSF assay buffer (20 mM Tris pH 8, 150 mM KCl, 2 mM CaCl$_2$, 0.005% wt/vol GDN, 0.0005% wt/vol CHS). SK2–4/CaM complex in DSF assay buffer was prepared at a concentration of 0.042 mg/ml. Compound 1 was dissolved in 100% DMSO at 10 mM before being diluted into the DSF assay buffer to 200 µM, with a final DMSO concentration of 2%. A negative control solution was prepared in the same way using only DMSO in the absence of compound. 7-Diethylamino-3-(4'-Maleimidylphenyl)–4-Methylcoumarin (CPM) was dissolved in 100% DMSO to a concentration of 4 mg/ml before being diluted into the DSF assay buffer to a concentration of 0.2 mg/ml.

Assay wells were prepared in triplicate by adding 30 µl of the SK2–4/CaM solution, 30 µl of the compound 1 solution or negative control solution, and 5 µl of the CPM solution together in Mx300P 96-well non-skirted plates (Agilent Technologies) such that each well contained 1.25 µg of SK2–4/CaM complex and 100 µM of compound 1, if present. The combined solution was briefly mixed followed by centrifugation. The plate was analyzed using Stratagene Mx3005P (Agilent Technologies) with the following method: temperature was held at 25°C for 30 min, followed by an increase of 0.5°C every 30 s until the temperature reached 95°C. Fluorescence readings were taken at each temperature.

Raw data were analyzed by fitting melt curves using an extended Boltzmann function to determine melting temperature $T_m$ for each replicate. Reported $\Delta T_m$ was determined by taking the mean of the three replicates for the compound 1 samples and subtracting the mean of the three replicates for the apo samples.

## Chemical synthesis of compound 1 (*N*-(benzo[c][1,2,5]oxadiazol-4-yl)-3-((4-methoxyphenyl)sulfonamido)benzamide) and compound 4 (*N*-(benzo[c][1,2,5]oxadiazol-4-yl)-4-(trifluoromethyl)benzamide)

### General methods

Unless otherwise noted, reagents and solvents were used as received from commercial suppliers. Compounds 2 and 3 were purchased from commercial suppliers. Proton NMR spectra were obtained on either a Bruker Avance spectrometer or a Varian Oxford 400 MHz spectrometer unless otherwise noted (*Figure 6—figure supplement 2*, *Figure 7—figure supplement 2*). NMR spectra are given in

ppm ($\delta$) and coupling constants, *J*, are reported in Hertz. Tetramethylsilane was used as an internal standard. Chemical shifts are reported in ppm relative to dimethyl sulfoxide ($\delta$ 2.50), methanol ($\delta$ 3.31), chloroform ($\delta$ 7.26), or other solvent as indicated in NMR spectral data. Peaks are reported as (s = singlet, d = doublet, t = triplet, q = quartet, m = multiplet or unresolved, br = broad signal, coupling constant(s) in Hz, integration). $^{13}$C NMR spectra were recorded with $^{1}$H-decoupling on Bruker AV 101 MHz spectrometers and are reported in ppm with the solvent resonance employed as the internal standard (DMSO-$d_6$ at 39.52 ppm). Mass spectra (ESI-MS) were collected using a Waters System (Acquity UPLC and a Micromass ZQ mass spectrometer) or Agilent-1260 Infinity (6120 Quadrupole); all masses reported are the *m/z* of the protonated parent ions unless recorded otherwise. The chemical names were generated using ChemDraw Professional v23 from Perkin Elmer Informatics. Temperatures are given in degrees Celsius. If not mentioned otherwise, all evaporations are performed under reduced pressure, typically between about 15 mm Hg and 100 mm Hg (=20–133 mbar).

**Chemical structure 1.** Synthetic scheme for N-(benzo[c][1,2,5]oxadiazol-4-yl)-3-((4-methoxyphenyl) sulfonamido) benzamide (1).

**Chemical structure 2.** Synthesis of methyl 3-((4-methoxyphenyl)sulfonamido)benzoate.

To a stirred solution of methyl 3-aminobenzoate (2.0 g, 1.0 equiv, 13.2 mmol) in DCM (10.0 ml) was added pyridine (3.14 g, 3.21 ml, 3 equiv, 39.7 mmol), followed by dropwise addition of 4-methoxybenzenesulfonyl chloride (3.28 g, 1.2 equiv, 15.6 mmol) at 25°C. Upon complete addition, the reaction mixture was stirred at RT for 2 hr. The progress of the reaction was monitored by TLC and LCMS. Upon completion of the reaction, the reaction mixture was diluted with water and extracted with DCM twice. The combined organic layers were washed with ice water twice, washed with brine, dried over Na$_2$SO$_4$, filtered, and concentrated under reduced pressure to obtain a residue. The crude residue was triturated with n-pentane and diethyl ether to get methyl 3-((4-methoxyphenyl)sulfonamido)benzoate (3.0 g, 9.1 mmol, 69% yield) as a brown solid.

## LCMS

System: Shimadzu–LCMS 2020 (single quad)
Column: ACQUITY UPLC BEH C18 1.7 µm, 2.1*50 mm
Column temperature: 40°C

Gradient (time (min.) / %B): 0.01/2, 0.08/2, 1.50/50, 2.20/98, 3.60/98, 4.20/2, 5.00/2
Eluent A: 0.1% HCOOH in water
Eluent B: 0.1% HCOOH in $CH_3CN$
Flow: 0.8 ml/min
**ESI-MS**, negative mode, $m/z$ 320.1 [M–H]⁻, Rt: 3.25 min, 99%.

## HPLC

Column: PRUDENT C18 150 X 4.6 mm, 5 µm
Flow: 1.0 ml/min
Mobile phase: (A) 0.01% TFA in water, (B) ACN
Gradient: T/%B 0/30, 1/70, 6/100, 8/100, 10/30, 12/30
Rt: 4.88 min, 98%
**¹H NMR** (400 MHz, DMSO-d6) $\delta$ = 10.42 (s, 1H), 7.71–7.68 (m, 3H), 7.63–7.57 (m, 1H), 7.41–7.35 (m, 2H), 7.10 – 7.0 (m, 2H), 3.79 (s, 3H), 3.76 (s, 3H).

**Chemical structure 3.** Synthesis of N-(benzo[c][1,2,5]oxadiazol-4-yl)-3-((4-methoxyphenyl)sulfonamido) benzamide (1).

To a stirred mixture of methyl 3-((4-methoxyphenyl)sulfonamido)benzoate (500 mg, 1.0 equiv, 1.56 mmol) in toluene (5 ml) was added triethylamine (315 mg, 434 µl, 2 equiv, 3.11 mmol) and benzo[c][1,2,5]oxadiazol-4-amine (252 mg, 1.2 equiv, 1.87 mmol) followed by dropwise addition of trimethylaluminum (2 M in hexane) (336 mg, 447 µl, 3 equiv, 4.67 mmol) at RT. After complete addition, the reaction mixture was heated at 80°C for 8 hr. The progress of the reaction was monitored by LCMS and TLC. The reaction mixture was cooled to RT and quenched with ice water and extracted with ethyl acetate twice. The combined organic layers were washed with brine, dried over $Na_2SO_4$, filtered, and concentrated to obtain a crude residue. The crude product was purified by normal phase silica gel chromatography, eluting with 10–25% ethyl acetate in hexane. The pure fractions were pooled and concentrated to obtain the product as a dark yellow gummy mass which was triturated with n-pentane and diethyl ether to afford N-(benzo[c][1,2,5]oxadiazol-4-yl)–3-((4-methoxyphenyl)sulfonamido) benzamide (**1**) (260 mg, 0.60 mmol, 39% yield) as a yellow solid.

## LCMS

System: Shimadzu–LCMS 2020 (single quad)
Column: ACQUITY UPLC BEH C18 1.7 µm, 2.1*50 mm
Column temperature: 40°C
Gradient (time (min.) / %B): 0.01/2, 0.08/2, 1.50/50, 2.20/98, 3.60/98, 4.20/2, 5.00/2
Eluent A: 0.1% HCOOH in water
Eluent B: 0.1% HCOOH in $CH_3CN$
Flow: 0.8 ml/min
**ESI-MS** $m/z$ 424.9, [M+H]⁺, Rt: 2.92 min, 99%

## HPLC

Column : X BRIDGE C18 150 X 4.6 mm, 3.5 µm

Flow: 1.0 ml/min

Mobile phase: (A) 0.1% formic acid in water and (B) ACN

Gradient: T/%B 0/5,1/5, 6/100, 8/100, 10/5, 12/5

Column temperature: 40°C

Rt: 7.03 min, 98.36%

**¹H NMR** (400 MHz, DMSO-$d_6$) $\delta$ 10.82 (s, 1H), 10.43 (s, 1H), 7.87 – 7.81 (m, 2H), 7.76 – 7.67 (m, 4H), 7.64 (dd, $J$ = 9.1, 7.1 Hz, 1 H), 7.46 – 7.39 (m, 1H), 7.37–7.32 (m, 1H), 7.10–7.03 (m, 2H), 3.79 (s, 3H) (*Figure 6—figure supplement 2A*).

**¹³C NMR** (101 MHz, DMSO-$d_6$) $\delta$ 165.6, 162.5, 149.6, 145.6, 138.4, 134.7, 133.5, 130.9, 129.3, 128.9, 126.7, 123.1, 123.0, 122.0, 119.5, 114.5, 111.8, 55.6 (*Figure 6—figure supplement 2B, C*).

**HRMS** (ESI) $m/z$ calculated for $C_{20}H_{17}N_4O_5S^+$ [M+H]⁺ 425.0914, found 425.0904.

**Chemical structure 4.** Synthetic scheme for N-(benzo[c][1,2,5]oxadiazol-4-yl)-4-(trifluoromethyl)benzamide (4).

## Synthesis of *N*-(benzo[*c*][1,2,5]oxadiazol-4-yl)-4-(trifluoromethyl)benzamide (4)

To a solution of benzo[*c*][1,2,5]oxadiazol-4-amine (51 mg, 0.38 mmol, 1.0 equiv) in THF (1.75 ml) in a 10-ml glass vial was added 4-(trifluoromethyl)benzoyl chloride (100 mg, 0.48 mmol, 1.25 equiv) followed by triethylamine (73 mg, 0.72 mmol, 1.9 equiv). The resulting mixture was shaken in a shaker for 18 hr. Then, the mixture was filtered and concentrated *in vacuo*. The residue was re-dissolved in methanol/acetonitrile (3 ml) and directly subjected to purification by HPLC. The product-containing fractions were pooled and concentrated *in vacuo* to furnish *N*-(benzo[*c*][1,2,5] oxadiazol-4-yl)–4-(trifluoromethyl)benzamide (4) (50 mg, 0.16 mmol, 43% yield) as a light yellow solid.

**¹H NMR** (400 MHz, DMSO-$d_6$) $\delta$ 11.15 (s, 1H), 8.23 – 8.18 (m, 2H), 7.98 – 7.93 (m, 3H), 7.88 – 7.85 (m, 1H), 7.67 (dd, $J$ = 9.1, 7.1 Hz, 1H) (*Figure 7—figure supplement 2A*).

**¹³C NMR** (101 MHz, DMSO-$d_6$) $\delta$ 165.2, 149.6, 145.6, 137.5, 133.5, 131.7 (q, $J$ = 32.2 Hz), 129.0, 126.4, 125.5 (q, $J$=3.7 Hz), 123.9 (d, $J$=272.8 Hz), 122.2, 112.1 (*Figure 7—figure supplement 2B, C*).

**¹⁹F NMR** (377 MHz, DMSO-$d_6$) $\delta$ –61.4 (*Figure 7—figure supplement 2D*).

**HRMS** (ESI) $m/z$ calculated for $C_{14}H_9F_3N_3O_2^+$ [M+H]⁺ 308.0641, found 308.0650.

# Additional information

### Competing interests

Samantha J Cassell, Weiyan Li, Simon Krautwald, Maryam Khoshouei, Yan Tony Lee, Joyce Hou, Wendy Guan, Stefan Peukert, Wilhelm Weihofen, Jonathan R Whicher: Employee of Novartis.

## Funding
No external funding was received for this work.

## Author contributions
Samantha J Cassell, Weiyan Li, Simon Krautwald, Conceptualization, Data curation, Formal analysis, Investigation, Methodology, Writing – original draft, Writing – review and editing; Maryam Khoshouei, Data curation, Formal analysis, Writing – original draft; Yan Tony Lee, Joyce Hou, Wendy Guan, Data curation; Stefan Peukert, Formal analysis, Investigation, Writing – original draft, Writing – review and editing; Wilhelm Weihofen, Conceptualization, Formal analysis, Investigation, Writing – original draft, Writing – review and editing; Jonathan R Whicher, Conceptualization, Data curation, Formal analysis, Investigation, Writing – original draft, Writing – review and editing

## Author ORCIDs
Samantha J Cassell https://orcid.org/0009-0003-3533-8482
Weiyan Li https://orcid.org/0009-0003-8743-4986
Simon Krautwald https://orcid.org/0009-0003-4090-2850
Maryam Khoshouei https://orcid.org/0000-0002-6039-772X
Wilhelm Weihofen https://orcid.org/0000-0002-7250-2398
Jonathan R Whicher https://orcid.org/0000-0002-3427-2501

Reviewer #3 (Public review): https://doi.org/10.7554/eLife.107733.3.sa1
Author response https://doi.org/10.7554/eLife.107733.3.sa2

---

# Additional files

## Supplementary files
MDAR checklist

1123451Supplementary file 1. Data collection parameters and refinement statistics.

## Data availability
The cryo-EM density maps for SK2-–4/CaM + $Ca^{2+}$ (EMD-70089), SK2-–4/CaM $Ca^{2+}$-free (EMD-70120), SK2-–4/CaM + apamin (EMD-70121), SK2-–4/CaM + compound 1 (EMD-70122), and SK2-–4/CaM + compound 4 (EMD-70145) have been deposited in the Electron Microscopy Data Bank (EMDB). The atomic coordinatedcoordinates for SK2-–4/CaM + $Ca^{2+}$ (9O48), SK2-–4/CaM $Ca^{2+}$-free (9O51), SK2-–4/CaM + apamin (9O52), SK2-–4/CaM + compound 1 (9O53), and SK2-–4/CaM + compound 4 (9O5O) have been deposited in the Protein Data Bank (PDB).

The following datasets were generated:

| Author(s) | Year | Dataset title | Dataset URL | Database and Identifier |
|---|---|---|---|---|
| Cassell SJ, Khoshouei M, Wilhelm WA, Whicher JR | 2025 | Cryo-EM structure of the human SK2-4 chimera/calmodulin channel complex in the Ca2+ bound state | https://www.ebi.ac.uk/emdb/EMD-70089 | Electron Microscopy Data Bank, EMD-70089 |
| Cassell SJ, Khoshouei M, Wilhelm WA, Whicher JR | 2025 | Cryo-EM structure of the human SK2-4 chimera/calmodulin channel complex in the Ca2+ free state | https://www.ebi.ac.uk/emdb/EMD-70120 | Electron Microscopy Data Bank, EMD-70120 |
| Cassell SJ, Khoshouei M, Wilhelm WA, Whicher JR | 2025 | Cryo-EM structure of the human SK2-4 chimera/calmodulin channel complex bound to the bee toxin apamin | https://www.ebi.ac.uk/emdb/EMD-70121 | Electron Microscopy Data Bank, EMD-70121 |

*Continued on next page*

*Continued*

| Author(s) | Year | Dataset title | Dataset URL | Database and Identifier |
|---|---|---|---|---|
| Cassell SJ, Khoshouei M, Wilhelm WA, Whicher JR | 2025 | Cryo-EM structure of the human SK2-4 chimera/calmodulin channel complex bound to a small molecule inhibitor | https://www.ebi.ac.uk/emdb/EMD-70122 | Electron Microscopy Data Bank, EMD-70122 |
| Cassell SJ, Khoshouei M, Wilhelm WA, Whicher JR | 2025 | Cryo-EM structure of the human SK2-4 chimera/calmodulin channel complex bound to a small molecule activator | https://www.ebi.ac.uk/emdb/EMD-70145 | Electron Microscopy Data Bank, EMD-70145 |
| Cassell SJ, Khoshouei M, Wilhelm WA, Whicher JR | 2025 | Cryo-EM structure of the human SK2-4 chimera/calmodulin channel complex in the Ca2+ bound state | https://doi.org/10.2210/pdb9O48/pdb | Worldwide Protein Data Bank, 10.2210/pdb9O48/pdb |
| Cassell SJ, Khoshouei M, Wilhelm WA, Whicher JR | 2025 | Cryo-EM structure of the human SK2-4 chimera/calmodulin channel complex in the Ca2+ free state | https://doi.org/10.2210/pdb9O51/pdb | Worldwide Protein Data Bank, 10.2210/pdb9O51/pdb |
| Cassell SJ, Khoshouei M, Wilhelm WA, Whicher JR | 2025 | Cryo-EM structure of the human SK2-4 chimera/calmodulin channel complex bound to the bee toxin apamin | https://doi.org/10.2210/pdb9O52/pdb | Worldwide Protein Data Bank, 10.2210/pdb9O52/pdb |
| Cassell SJ, Khoshouei M, Wilhelm WA, Whicher JR | 2025 | Cryo-EM structure of the human SK2-4 chimera/calmodulin channel complex bound to a small molecule inhibitor | https://doi.org/10.2210/pdb9O53/pdb | Worldwide Protein Data Bank, 10.2210/pdb9O53/pdb |
| Cassell SJ, Khoshouei M, Wilhelm WA, Whicher JR | 2025 | Cryo-EM structure of the human SK2-4 chimera/calmodulin channel complex bound to a small molecule activator | https://doi.org/10.2210/pdb9O5O/pdb | Worldwide Protein Data Bank, 10.2210/pdb9O5O/pdb |

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
