## [Editor Report · eLife Assessment]

In this **important** manuscript, Cassell and colleagues set out on a mechanistic and pharmacological exploration of an engineered chimeric small conductance calcium-activated potassium channel 2 (SK2). They show **compelling** evidence that the SK2 channel possesses a unique extracellular structure that modulates the conductivity of the selectivity filter, and that this structure is the target for the SK2 inhibitor apamin. The interpretations are sound and the writing is clear, and the manuscript was strengthened during review by providing more detailed information for the electrophysiological experiments and the structural analyses attempted, in addition to relating dilation of the filter to mechanisms of inactivation in other potassium channels. This high-quality study will be of interest to membrane protein structural biologists, ion channel biophysicists, and chemical biologists, and will help to inform future drug development targeting SK channels.

---

## [Referee Report · Reviewer #3 (Public review)]

This is a fundamentally important study presenting cryo-EM structures of a human small conductance calcium-activated potassium (SK2) channel in the absence and presence of calcium, or with interesting pharmacological probes bound, including the bee toxin apamin, a small molecule inhibitor, and a small molecule activator. As efforts to solve structures of the wild-type hSK2 channel were unsuccessful, the authors engineered a chimera containing the intracellular domain of the SK4 channel, the subtype of SK channel that was successfully solved in a previous study (reference 13). The authors present many new and exciting findings, including opening of an internal gate (similar to SK4), for the first time resolving the S3-S4 linker sitting atop the outer vestibule of the pore and unanticipated plasticity of the ion selectivity filter, and the binding sites for apamin, one new small molecule inhibitor and another small molecule activator. Appropriate functional data are provided to frame interpretations arising from the structures of the chimeric protein; the data are compelling, the interpretations are sound, and the writing is clear. This high-quality study will be of interest to membrane protein structural biologists, ion channel biophysicists, and chemical biologists, and will be valuable for future drug development targeting SK channels.

Comments on revisions:

The authors have done a nice job of revising the manuscript to address the issues raised in the first round of review and I have no further suggestions.

---

## [Author Response]

The following is the authors’ response to the original reviews.

**Reviewer #1 (Public review):**
The small conductance calcium-activated potassium channel 2 (SK2) is an important drug target for treating neurological and cardiovascular diseases. However, structural information on this subtype of SK channels has been lacking, and it has been diOicult to draw conclusions about activator and inhibitor binding and action in the absence of structural information.Here the authors set out to (1) determine the structure of the transmembrane regions of a mammalian SK2 channel, (2) determine the binding site of apamin, a historically important SK2 inhibitor whose mode of action is unclear, and (3) use the structural information to generate a novel set of activators/inhibitors that selectively target SK2.The authors largely achieved all the proposed goals, and they present their data clearly.Unable to solve the structure of the human SK2 due to excessive heterogeneity in its cytoplasmic regions, the authors create a chimeric construct using SK4, whose structure was previously solved, and use it for structural studies. The data reveal a unique extracellular structure formed by the S2-S3 loop, which appears to directly interact with the selectivity filter and modulate its conductivity. Structures of SK2 in the absence and presence of the activating Ca2+ ions both possess non-K+-selective/conductive selectivity filters, where only sites 3 and 4 are preserved. The S6 gates are captured in closed and open states, respectively. Apamine binds to the S2-S3 loop, and unexpectedly, induces a K+ selective/conductive conformation of the selectivity filter while closing the S6 gate.Through high-throughput screening of small compound libraries and compound optimization, the group identified a reasonably selective inhibitor and a related compound that acts as an activator. The characterization shows that these compounds bind in a novel binding site. Interestingly, the inhibitor, despite binding in a site diOerent from that of apamine, also induces a K+ selective/conductive conformation of the selectivity filter while the activator induces a non-K+ selective/conductive conformation and an open S6 gate.The data suggest that the selectivity filter and the S6 gate are rarely open at the same time, and the authors hypothesize that this might be the underlying reason for the small conductance of SK2. The data will be valuable for understanding the mechanism of SK2 channel (and other SK subtypes).Overall, the data is of good quality and supports the claims made by the authors. However, a deeper analysis of the cryo-EM data sets might yield some important insights, i.e., about the relationship between the conformation of the selectivity filter and the opening of the S6 gate.

We attempted focused 3D classification to identify subsets of particles with the S6 open and the SF in a conductive state but were not able to isolate such a particle class. This indicates that either none or a very small percentage of particles exists in a fully conductive state. This sentence was included in the results section:

“Focused 3D classification of the S3-S4 linker was unsuccessful in identifying particles subsets with a dilated extracellular constriction suggesting that either none or a very small percentage of Ca^2+^-bound SK2-4 is in a conductive state”

Some insight and discussion about the allosteric networks between the SF and the S6 gate would also be a valuable addition.

The extracellular constriction is in the same non-conductive conformation in the Ca^2+^ bound and Ca^2+^ -free SK2-4 structures suggesting that the conformation of S3-S4 linker/SF and the S6 are not allosterically coupled. We predict that Ca^2+^ opens the intracellular gate and another physiological factor (not yet identified) promotes extracellular gate opening. These sentences were added to the results and discussion: “This along with the similar conformation of the S3-S4 linker in the Ca^2+^ -bound and Ca^2+^ -free states of SK2-4 suggest that Ca^2+^ -dependent intracellular gate dynamics are not coupled to the conformation of the S3-S4 linker. Other yet to be identified physiological factors may be required to dilate the extracellular constriction.”

“Alternatively, other physiological factors, such as PIP2[46,47] or protein-protein interactions[48-50], may exist in live cells that modulate the interaction between S3-S4 linker and the selectivity filter.”

**Reviewer #2 (Public review):**
Summary:The authors have used single-particle cryoEM imaging to determine how small-molecule regulators of the SK channel interact with it and modulate their function.Strengths:The reconstructions are of high quality, and the structural details are well described.Weaknesses:The electrophysiological data are poorly described. Several details of the structural observations require a mechanistic context, perhaps better relating them to what is known about SK channels or other K channel gating dynamics.

As recommended, additional details for electrophysiological data were added to the results, methods, and figure legends for clarification.

The most pressing point I have to make, which could help improve the manuscript, relates to the selectivity filter (SF) conformation. Whether the two ion-bound state of SK2-4 (Figure 4A) represents a non-selective, conductive SF occluded by F243 or represents a C-type inactivated SF, further occluded by F243, is unclear. It would be important to discuss this. Reconstructions of Kv1.3 channels also feature a similar configuration, which has been correlated to its accelerated C-type inactivation.

Structural overlays of Ca^2+^ bound SK2-4, HCN, and C-type inactivated Kv1.3 selectivity filters demonstrate that each have conformational diVerences and it is diVicult to definitively determine if the SK2-4 selectivity filter is in a non-selective conformation like HCN or a C-type inactivated conformation like Kv1.3. Based on the number of ions observed in the filter and the position of Tyr361 we believe the selectivity filter most closely resembles that of HCN. Importantly, the selectivity filter conformation observed in the SK2-4 Ca^2+^ -bound and Ca^2+^ -free structures is ultimately nonconductive due to the Phe243 extracellular constriction blocking K^+^ eVlux.

A comparison of the SK2-4 selectivity filter to HCN and C-type inactivated Kv1.3 was included in Figure 4 and this sentence was included in the results section:

“The selectivity filter of SK2-4 resembles that of to HCN in both the position of Tyr361 and the number of K^+^ coordination sites (Fig 4E,F,G,H)”

Furthermore, binding of a toxin derivative to Kv1.3 restores the SF into a conductive form, though occluded by the toxin. It appears that apamin binding to SK2-4 might be doing something similar. Although I am not sure whether SK channels undergo C-type inactivation like gating, classical MTS accessibility studies have suggested that dynamics of the SF might play a role in the gating of SK channels. It would be really useful (if not essential) to discuss the SF dynamics observed in the study and relate them better to aspects of gating reported in the literature.

Extracellular toxin binding to SK2-4 and K_v_1.3 induce a conformational change in the selectivity filter to produce a canonical K^+^ selective structure with four coordination sites. However, the mechanism by which the toxins produce the conformational change is diVerent. For SK2-4, apamin interacts primarily with S3-S4 linker residues and induces a shift in the S3-S4 linker away from the pore axis. This in turn prevents the hydrogen bonds between Arg240 and Tyr245 of the S3-S4 linker and Asp363 at the C-terminus of the selectivity filter to produce a selectivity filter conformation with four K^+^ coordination sites. For K_v_1.3, the sea anemone toxin ShK binds directly to the C-terminus of the selectivity filter disrupting interactions required for the C-type inactivated structure and thereby inducing the conformational change. These sentences were added to the results:

“Toxin induced selectivity filter conformational change has also been reported for K_v</sub 1.3 with the sea anemone toxin ShK. However, unlike apamin binding to SK2-4, ShK binds directly to the Kv 1.3 selectivity filter to convert a C-type inactivated conformation to a canonical K^+^ selective structure with four coordination sites [39,40]. The change in selectivity filter conformation in apamin-bound SK2-4 seems to be driven instead by the weakening of interactions between the selectivity filter and the S3-S4 linker.”_

The SF of K channels, in conductive states, are usually stabilized by an H-bond network involving water molecules bridged to residues behind the SF (D363 in the down-flipped conformation and Y361). Considering the high quality of the reconstructions, I would suspect that the authors might observe speckles of density (possibly in their sharpened map) at these sites, which overlap with water molecules identified in high-resolution X-ray structures of KcsA, MthK, NaK, NaK2K, etc. It could be useful to inspect this region of the density map.

We did not observe strong density near Y361 or D363 that could be confidently model as water. However, in the structures of SK2-4 bound to apamin and compound 1 Tyr361 in the selectivity filter rotates 180° and forms a hydrogen bond with Thr355 in the pore helix. The homologous hydrogen bond is also observed in SK4 and the conductive/ K^+^ selective selectivity filter conformation of Kv1.3. The rotation of Tyr361 to form a hydrogen bond with Thr355, reorientation of Asp363 and Trp350 into hydrogen bonding position, and the presence of four K^+^ coordination sites upon binding of apamin and compound 1 strongly suggest that the selectivity filter is in a K^+^ selective/conductive conformation. The Tyr361/Thr355 hydrogen bond is now described in the paper and shown in Figures 4D, 5D, and S6F.

**Reviewer #3 (Public review):**
This is a fundamentally important study presenting cryo-EM structures of a human small conductance calcium-activated potassium (SK2) channel in the absence and presence of calcium, or with interesting pharmacological probes bound, including the bee toxin apamin, a small molecule inhibitor, and a small molecule activator. As eOorts to solve structures of the wild-type hSK2 channel were unsuccessful, the authors engineered a chimera containing the intracellular domain of the SK4 channel, the subtype of SK channel that was successfully solved in a previous study (reference 13). The authors present many new and exciting findings, including opening of an internal gate (similar to SK4), for the first time resolving the S3-S4 linker sitting atop the outer vestibule of the pore and unanticipated plasticity of the ion selectivity filter, and the binding sites for apamin, one new small molecule inhibitor and another small molecule activator. Appropriate functional data are provided to frame interpretations arising from the structures of the chimeric protein; the data are compelling, the interpretations are sound, and the writing is clear. This high-quality study will be of interest to membrane protein structural biologists, ion channel biophysicists, and chemical biologists, and will be valuable for future drug development targeting SK channels.The following are suggestions for strengthening an already very strong and solid manuscript:(1) It would be good to include some information in the text of the results section about the method and configuration used to obtain electrophysiological data and the limitations. It is not until later in the text that the Qube instrument is mentioned in the results section, and it is not until the methods section that the reader learns it was used to obtain all the electrophysiological data. Even there, it is not explicitly mentioned that a series of diOerent internal solutions were used in each cell where the free calcium concentration was varied to obtain the data in Figure1C. Also, please state the concentration of free calcium for the data in Figure 1B.

As recommended, additional details for electrophysiological data were added to the results, methods, and figure legends for clarification.

(2) The authors do a nice job of discussing the conformations of the selectivity filter they observed here in SK as they relate to previous work on NaK and HCN, but from my perspective the authors are missing an opportunity to point out even more striking relationships with slow C-type inactivation of the selectivity filter in Shaker and Kv1 channels. C-type inactivation of the filter in Shaker was seen in 150 mM K using the W434F mutant (PMC8932672) or in 4 mM K for the WT channel (PMC8932672), and similar results have been reported for Kv1.2 (PMC9032944; PMC11825129) and for Kv1.3 (PMC9253088; PMC8812516) channels. For Kv1.3, C-type inactivation occurs even in 150 mM K (PMC9253088; PMC8812516). Not unlike what is seen here with apamin, binding of the sea anemone toxin (ShK) with a Fab attached (or the related dalazatide) inserts a Lys into the selectivity filter and stabilizes the conducting conformation of Kv1.3 even though the Lys depletes occupancy of S1 by potassium (PMC9253088; PMC8812516). Or might the conformation of the filter be controlled by regulatory processes in SK2 channels? I think connecting the dots here would enhance the impact of this study, even if it remains relatively speculative.

Please see the response to reviewer 2’s comments for a comparison of the selectivity filter structure between SK2-4 and C-type inactivated K_v_1.3 and a discussion of toxin induced selectivity filter conformational change.

What is known about how the functional properties of SK2 channels (where the filter changes conformation) diOer from SK4, where the filter remains conducting (reference 13)? Is there any evidence that SK2 channels inactivate?

Compared with SK4, SK2 has some unique properties such as lower conductance and the ability to switch between low- and high-open probability states. Mutation of Phe243 suggests that the S3-S4 linker conformation contributes to the low conductance. This is included in the discussion.

“Such a mechanism may explain some properties of SK2 that are not observed in SK4, which lacks an S3-S4 linker, such as its low conductance (~10 pS) and the ability to switch between low- and high-open probability states[3,4]. Indeed, mutation of Phe243 in rat SK2 produced a 2-fold increase in channel conductance[5].”

Or might the conformation of the filter be controlled by regulatory processes in SK2 channels? I think connecting the dots here would enhance the impact of this study, even if it remains relatively speculative.

Please see the response to reviewer 1’s comments for a discussion of the potential physiological role of the S3-S4 linker/extracellular constriction and its mechanism for opening.

**Reviewer #1 (Recommendations for the authors):**
I enjoyed reading your paper and am intrigued by your findings on the selectivity filter of SK2. I've got a few recommendations for data analysis and a couple of questions that might contribute to the discussion.In your Ca2+-bound dataset, have you tried to parse out any alternative conformations (e.g., by using 3D classification, or 3D variability)? Do you think there might be a small(er) population of particles that adopt a fully open conformation? If you haven't done this already, I would recommend doing so. You have a rather large number of particles in your final 3D reconstruction (~660k), so there might be some hidden conformations that could contribute to our understanding of the system.I would recommend doing the same for your compound 4-bound data set.

Please see above for response to this recommendation.

Do you think apamine works solely as a pore blocker, or does its binding perhaps also aOect the S6 gate via allosteric networks (perhaps the same ones that induce the formation of the K+ conductive SF through binding of compound 1 above the S6 gate?)?

Apamin binding does not change the conformation of the pore helices (S5 or S6) and thus we believe it acts primarily as a pore blocker. The following was added to the results section:

“Overall, the apamin-bound SK2-4/CaM structure resembles Ca^2+^-bound SK2-4. The Nterminal lobe of CaM engages with the S_45_ A helix, the S5 and S6 helices adopt a similar conformation, and the intracellular gate Val390 is open with a radius of 3.5 Å (Fig 2D). The most significant conformational change is in the position of the S3-S4 linker, which shifts ~2 Å away from the pore axis to accommodate apamin binding.”

Is there a mechanistic explanation for why it might be diOicult/energetically costly for the SF to be conductive and the S6 gate to be open at the same time?

Not to our knowledge.

I also have these minor recommendations:-In all figures showing density, include the threshold/sigma value at which density is shown.-For all ligands and ions, include half-map data.

Sigma values were added for all figures legends displaying cryoEM density. The displayed maps are the sharpened full maps.

**Reviewer #2 (Recommendations for the authors):**
Is it possible to provide a structure-sequence guided explanation for the diOerent aOinity of compound 1 for SK2 vs SK4?

Yes. The following is now included in the results section and a panel was added to Figure S6D.

“However, for SK4 Thr212 replaces SK2 Ser318 and Trp216 (homologous to SK2 Trp322) is conserved but adopts a diVerent rotamer conformation (Fig S6D). Both changes occlude the compound 1 binding site in SK4 and would likely reduce compound 1 potency on SK4 as observed in the functional data.”

Is it possible to propose a model of modulation by compound 1/4 where the authors can comment on the conformational dependence of compound binding? That is, do they bind exclusively to the identified conformational states of the channel, or are they able to bind to both closed and open channels, but bias one state over the other?

The clash between compound 1 and Thr386 in the open conformation of the S6 helices suggests that compound 1 would preferentially bind to closed state of SK2. Similarly, the clash between compound 4 and Ile380 in the closed conformation of the S6 helices suggests that compound 4 would preferentially bind to the open state of SK2. This was included in the discussion:

“This proposed mechanism of modulation suggests that compound 1 may bind preferentially to the closed conformation of the S6 helices and compound 4 may bind preferentially to the open conformation of the S6 helices.”

Please provide the calcium concentration used to generate the data in Figure 1B. The calcium concentration is now stated in the legend for Fig 1B:

“Intracellular solution contains 2 µM Ca^2+^ based on calculation using Maxchelator (see methods)”

Essential and critically important descriptions of experiments in Figure 7A are lacking. It would be essential to describe properly, with care, what the currents and the conditions of measurements are. If these currents are obtained by subtracting leak currents by adding other drugs, it would be good to comment on whether the latter compete with compounds 1/4.

As recommended, additional details for electrophysiological data were added to the results, methods, and figure legends for clarification. SK currents were obtained by subtracting leak currents by adding UCL1684 only at the end of experiments. UCL1684 is not expected to interfere with eVect of compound 1 or 4 given diVerent binding sites and mechanisms.

If Compound 1 changes the structure of the SF (Figure 6F), would it also promote apamin binding? Given that both these agents produce a similar change in the SF, could each favor the binding of the other?

Since apamin binds to the S3-S4 linker it is unlikely that the selectivity filter conformational change observed in the compound 1 bound structure would aVect apamin binding.